# LRRK2 interactions with microtubules are independent of LRRK2-mediated Rab phosphorylation

Tuyana Malankhanova [ID], Zhiyong Liu, Enquan Xu, Nicole Bryant [ID], Ki Woon Sung [ID], Huizhong Li, Samuel Strader [ID] & Andrew B West [ID] ✉

## Abstract

**Deregulated microtubules are common defects associated with neurodegenerative diseases. Recent cryo-electron microscopy studies in cell lines overexpressing Parkinson's disease-associated LRRK2 suggest microtubule surfaces may regulate kinase activity by stabilizing different LRRK2 conformations. In macrophages with high endogenous LRRK2 expression, we find that nocodazole treatment destabilizes microtubules and impairs LRRK2-mediated Rab phosphorylation. GTP supplementation restores nocodazole-reduced Rab phosphorylation, linking LRRK2 kinase action to cellular GTP levels. Chemical microtubule stabilization, and kinetically trapping LRRK2 to microtubule surfaces, has negligible effects on Rab phosphorylation. In contrast, trapping LRRK2 to LAMP1-positive membranes upregulates LRRK2-mediated Rab phosphorylation. Proximity-labeling proteomics and colocalization studies show that LRRK2 robustly interacts with both polymerized and free tubulin transiently and independently of LRRK2 kinase activity. Endogenous LRRK2 complexed with type I inhibitors in neurons and macrophages fails to stably interact with microtubules, whereas bulky N-terminal tags fused to LRRK2 promotes stable microtubule binding in cell lines. Collectively, these results show that tubulin isoforms and microtubules are transient LRRK2-interacting proteins non-essential for LRRK2-mediated Rab phosphorylation.**

**Keywords** Cell Signaling; Cytoskeleton; Parkinson's Disease; Protein Kinase; GTPase
**Subject Categories** Cell Adhesion, Polarity & Cytoskeleton; Membranes & Trafficking; Signal Transduction

## Introduction

The *LRRK2* gene encodes the LRRK2 protein, a 288 kDa serine/threonine kinase and GTPase implicated in Parkinson's disease (PD). *LRRK2* is also linked to other diseases involving inflammation and innate immune responses that include Crohn's disease and mycobacterial infections (Wallings and Tansey, 2019). Pathogenic LRRK2 mutations that cause PD can occur in the ROC (Ras in complex) domain, COR (C-terminal of Roc) domain, and kinase domain, facilitating an upregulation of LRRK2 kinase activity (Alessi and Pfeffer, 2024; West et al, 2007). Several type I LRRK2 kinase inhibitors that bind to the LRRK2 ATP pocket and inhibit LRRK2-mediated Rab phosphorylation have advanced to clinical trials in PD (West and Schwarzschild, 2023). Other domains encoded in the LRRK2 protein include a leucine-rich repeat region and a WD-40-like domain, with proposed roles in mediating enzymatically active conformations of LRRK2 and interactions with co-factors (Gilsbach and Kortholt, 2014). The LRRK2 ROC domain is structurally and functionally similar to the Rab subfamily of Ras small GTPases (Bosgraaf and Van Haastert, 2003), and LRRK2 phosphorylates switch II effector-loops of several other small Rab substrates including Rab8a, Rab10, Rab12, and Rab35 (Steger et al, 2016). Although LRRK2 can phosphorylate GDP-bound GTPases in vitro, in cells, LRRK2-mediated Rab phosphorylation appears restricted to membrane-bound and activated GTP-bound Rab substrates (Liu et al, 2018; Gomez et al, 2019). N-terminal repeat domains in LRRK2 can bind and interact with Rab protein substrates (McGrath et al, 2021; Vides et al, 2022; Pettersen et al, 2021), and recent reports show that molecular trapping of transfected LRRK2 to membranes in HEK293T cells promotes LRRK2-mediated Rab phosphorylation (Kluss et al, 2022b). Initial reports also suggest the accumulation of phosphorylated Rab proteins in tissues and biofluids in PD, even in individuals without pathogenic *LRRK2* mutations (Wang et al, 2022; Yan et al, 2018; Yuan et al, 2024). Factors regulating endogenous LRRK2 activity are therefore of biological and therapeutic interest.

LRRK2 protein is largely soluble throughout the cytoplasm, but some LRRK2 protein localizes to vesicles and other membranes in cells, as well as to the outer surface of microtubules (Lee et al, 2010; Biskup et al, 2006). LRRK2 protein interacts with different ß-tubulin isoforms and dynamic microtubules in dividing cells (Law et al, 2014). Tubulin dimers each have two GTP-binding sites, an exchangeable site on the ß-subunit and a non-exchangeable site on the α-subunit. During or after polymerization, GTP on the ß-subunit is hydrolyzed to GDP and then sequestered along the tubules (Hyman et al, 1992). The small molecule nocodazole is thought to impair the addition of tubulin to the growing

Duke Center for Neurodegeneration Research, Department of Pharmacology and Cancer Biology, Duke University, Durham, NC, USA. ✉E-mail: andrew.west@duke.edu

microtubule ends (Vasquez et al, 1997). Microtubules are crucial to neuronal and myeloid cell function, contributing to vesicle trafficking and scaffolding protein organization, among other functions. Microtubule dysregulation is strongly associated with accumulation of the microtubule-associated protein tau in tauopathies (Dubey et al, 2015), and most LRRK2 cases with pathogenic mutations harbor Alzheimer's type tau pathology (Henderson et al, 2019).

Past reports show that transfected N-terminal tagged LRRK2 protein in cell lines with the R1441C, R1441G, Y1699C, or I2020T *LRRK2* pathogenic mutations increase (exogenous) LRRK2 interactions with microtubules, aligning LRRK2 protein along regular periodic microtubule intervals (Kett et al, 2011). In contrast, WT-LRRK2 and G2019S-LRRK2 proteins failed to demonstrate similar microtubule binding patterns. However, type I LRRK2 kinase inhibitor binding to the exogenous protein-tagged (i.e., transfected) WT or G2019S-LRRK2 strongly stabilizes interactions between LRRK2 and microtubules (Deniston et al, 2020). Capitalizing on the apparent stable interaction between over-expressed LRRK2 and mature microtubules, a cryo-electron tomography approach was able to resolve high-resolution structures for N-terminal YFP-I2020T-mutated LRRK2 protein wrapped around microtubules (Watanabe et al, 2020). Work with a C-terminal fragment of LRRK2 spanning the catalytic domains demonstrates that type I inhibitor binding and microtubule association stabilizes a kinase-active (closed) conformation of LRRK2 (Snead et al, 2022). The transfected chimeric LRRK2 COR and WD-40-like domains roll around the microtubule in a way described to disrupt the transport of membranous cargo (Deniston et al, 2020). Tubulin is very abundant in cells and tissues, for example, composing 10% of total protein in brain tissue (Verdier-Pinard et al, 2009). Whether or not microtubules play an important role in mediating endogenous LRRK2 activity and Rab substrate phosphorylation, especially under physiological conditions, has not been addressed, but could plausibly represent one way that LRRK2 activity may be deregulated in ongoing neurodegeneration and microtubule dysfunction occurring in disease.

Macrophages are important non-ciliated cells in the innate immune system known to express high endogenous levels of both LRRK2 protein and Rab protein substrates, especially Rab10 (Liu et al, 2020). Further, macrophages have abundant and dynamic microtubules known to facilitate both signaling and motility processes. Through multiple lines of experimental evidence presented here, we find that tubulin or microtubule interactions do not significantly regulate LRRK2 activity and Rab phosphorylation, nor does the forced recruitment of LRRK2 to microtubule surfaces. Impaired Rab phosphorylation via nocodazole exposures can be rescued through supplementation of cellular GTP, implicating dynamic GTP in the maintenance of dynamic phosphorylated Rab protein levels. Endogenous LRRK2 protein does not stably interact with microtubules in vivo, even when LRRK2 is bound to type I inhibitors, though LRRK2 prominently associates (albeit in a kinase-independent manner) with different tubulin isoforms and microtubules in cells. These findings suggest caution may be warranted with the interpretation of non-native LRRK2 protein complexed with microtubules as these complexes appear to rely on non-physiological stoichiometries between LRRK2 and tubulins. Further, transient LRRK2 interactions with microtubules do not appear to impart an important regulatory

aspect in mediating phosphorylated Rab levels. In contrast, Rab membrane localization and LRRK2 recruitment to membranes appear critically important in the regulation of phosphorylated Rabs, irrespective of membrane damage or stress, or microtubule polymerization state. The identification of additional factors that regulate LRRK2 and Rab interactions at the membrane in macrophages and other cells with higher endogenous LRRK2 expression may provide additional insight into the molecular underpinnings of LRRK2 kinase regulation in both health and disease.

# Results

## Nocodazole treatment blocks LRRK2-mediated Rab phosphorylation

Recent studies suggest microtubules might regulate LRRK2 kinase activity by promoting the stabilization of a kinase-active LRRK2 conformation (Snead et al, 2022; Watanabe et al, 2020). These and other past studies have been primarily performed in transfected cells, or with recombinant proteins in vitro that may not mimic physiologically relevant stoichiometries of protein expression. To investigate further how microtubule dynamics might regulate LRRK2 activity and Rab10 phosphorylation in macrophages, we first selected docetaxel and nocodazole as well-described chemicals that either promote or destabilize microtubules in macrophages (Steinmetz and Prota, 2018). Mouse bone marrow-derived macrophages (BMDMs) were cultured from WT-LRRK2 and G2019S-LRRK2 BAC mice that stably express *LRRK2* higher than mouse *Lrrk2* but at levels similar to that observed in human primary cultured macrophages (Xu et al, 2022). Matched cells from *Lrrk2* knockout mice were also cultured as controls for LRRK2 antibody specificity in immunoblots, immunocytochemistry and immunohistochemistry approaches utilized across this study.

Treatment of macrophages expressing WT-LRRK2 or G2019S-LRRK2 showed that nocodazole potently inhibited the ratio of pT73-Rab10 (later referred to as pRab10) to total Rab10 (Fig. 1A–C). In contrast, docetaxel treatment had no effect on pRab10 levels. To help ensure relative linearity of measurements of proteins like LRRK2 and pRab10 from immunoblots in this study, all signals measured were acquired digitally in non-saturated immunoblot exposures that maintained good linearity across detection ranges (Fig. EV1A,B). Accordingly, none of the drugs had significant effects on total LRRK2 protein, total Rab10, or tubulin levels in the time frames of drug exposures (Figs. 1A,B and EV1C,D). Immunocytochemistry evaluation of macrophages fixed with paraformaldehyde and treated with saponin (to help eliminate freely soluble proteins from detection) confirmed that microtubules rapidly destabilized throughout the macrophage cytoplasm after nocodazole treatment, whereas docetaxel stabilized microtubules (Fig. 1D). Nocodazole treatment also reduced total Rab10 immunocytochemistry intensity in cells (Fig. 1D,E), whereas docetaxel had no significant effects. Vehicle treatments consisted of 0.01% DMSO.

To confirm the effects of nocodazole on pRab10 inhibition in human cells with high endogenous LRRK2 and Rab10 expression, the A549 lung cell line and human primary blood monocyte-derived macrophages (MDMs) were selected and treated with either docetaxel,

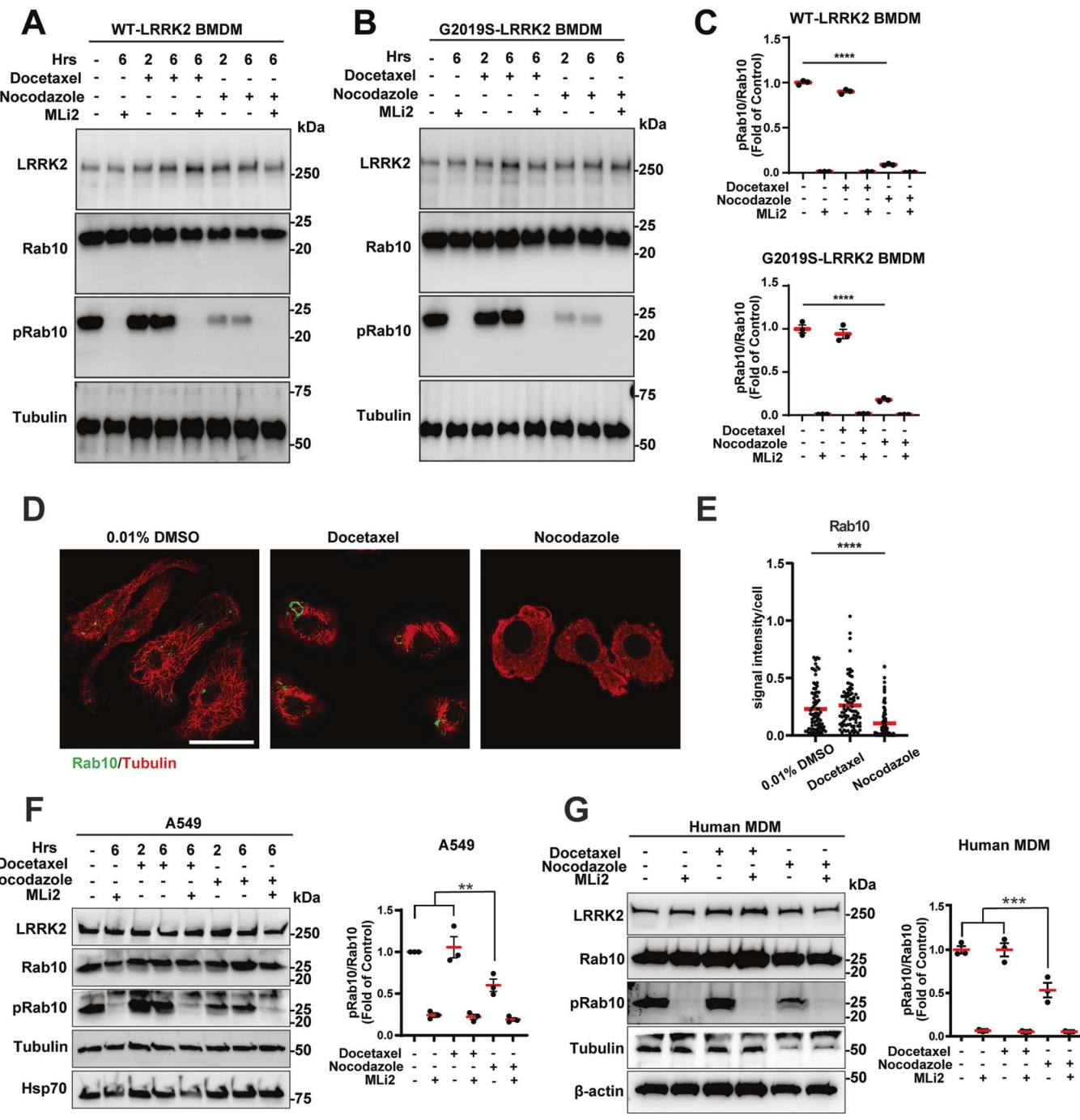

**Figure 1. Nocodazole potently blocks LRRK2-mediated Rab phosphorylation.**

(A, B) Representative immunoblots of WT-LRRK2 and G2019S-LRRK2 BMDM lysates. Cells were treated with docetaxel (10 μM), nocodazole (10 μM), and MLi2 (250 nM), for the time indicated prior to lysis. (C) Relative (fold of non-treated cells) quantification of the ratio of pRab10 to total Rab10 is shown, where each dot represents immunoblot analysis of one biological replicate ($n = 3$ biologically independent experiments). Group means are shown. Error bars represent ± SEM. Significance was assessed by one-way ANOVA with Tukey's *post hoc* test, with **** representing $P < 0.0001$. (D, E) Representative immunocytochemistry of total Rab10 (green) and tubulin (red) from BMDMs (WT-LRRK2) treated with docetaxel or nocodazole (10 μM, 2 h), then paraformaldehyde-fixed and saponin-treated. Scale bar: 10 μm. Quantification of total Rab10 intracellular signal intensity in BMDMs is shown, determined from high-resolution confocal-reconstructed images, where each dot represents the analysis of Rab10 intensity in one cell, with ≥20 cells analyzed each from three biological replicates. Group means are shown. Error bars represent ± SEM. Significance was assessed by one-way ANOVA with Tukey's post hoc test, with **** representing $P < 0.0001$. (F, G) A549 cells and human MDMs were treated and analyzed for the ratio of pRab10 to total Rab10 in an identical manner to BMDMs in (A–C). Each dot in the graphs represents the analysis of one biological replicate ($n = 3$ biologically independent experiments). Group means are shown. Error bars represent ± SEM. Significance was assessed by one-way ANOVA with Tukey's post hoc test, with ** representing $P < 0.01$, *** representing $P < 0.001$. Exact $P$ values: $P = 0.0074$ (F); $P = 0.0003$ (G).

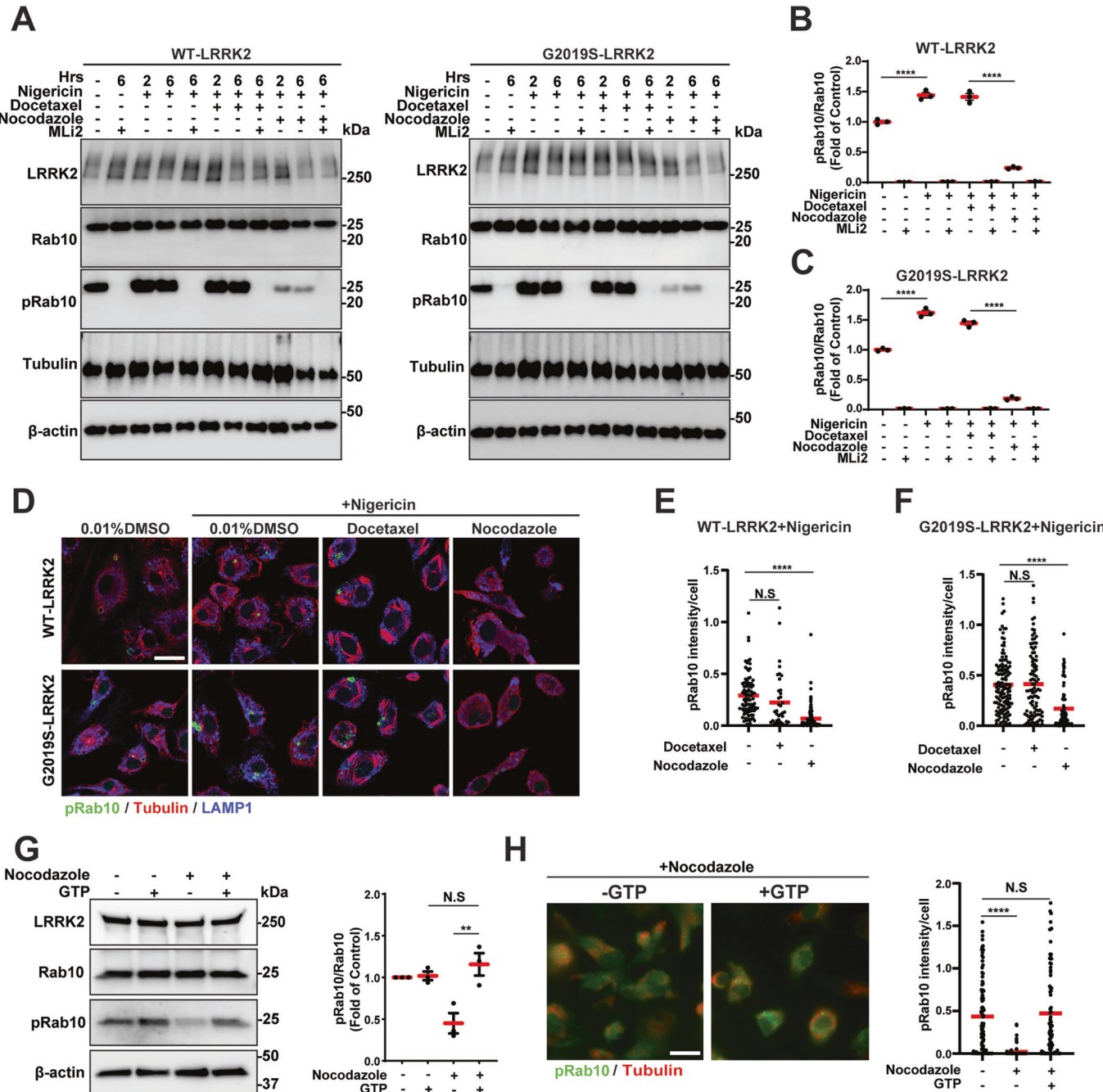

nocodazole, or the LRRK2 kinase inhibitor MLi2. The ratio of pRab10 to total Rab10, as determined by immunoblotting (Fig. 1F,G), was reduced with nocodazole, but not docetaxel (Fig. 1F,G). Nocodazole treatment further caused a partial reduction of endogenous pRab10 levels in HEK293T cells transfected with LRRK2 or LRRK2 with pathogenic mutations G2019S, Y1699C, or R1441C (Fig. EV2). Thus, in macrophages of human or mouse origins, as well as in different human cell lines, nocodazole reduces LRRK2-mediated pRab10 levels. In contrast, microtubule stabilization via docetaxel treatment has no effect on pRab10 levels.

Since the phosphorylation levels of Rab10 are mediated in part through the PPM1H phosphatase, we sought to determine whether

nocodazole might affect the ratio of pRab10 to total Rab10 through upregulating PPM1H activity. In macrophages, MLi2 treatment results in the loss of pRab10 within minutes. However, the presence of nocodazole did not hasten MLi2-mediated loss of pRab10 (Fig. EV3). These results suggest that nocodazole may not be affecting pRab10 levels through enhanced PPM1H activity. LRRK2 relocalization along microtubules has been suggested as a key regulatory event for LRRK2 kinase activity (Deniston et al, 2020; Watanabe et al, 2020). High-resolutions structural analysis of the LRRK2-microtubule complex suggests possible occlusion of otherwise solvent-oriented antibody-binding epitopes. Recent studies demonstrate that the N-terminal LRRK2 antibody UDD3 binds to

**Figure 2. Nocodazole blocks LRRK2 activation with lysosomal stress in a GTP-dependent manner.**

(A) Representative immunoblots of WT-LRRK2 and G2019S-LRRK2 BMDM lysates. Lysosomal stress was induced with Nigericin (5 μM) as indicated, with docetaxel (10 μM), nocodazole (10 μM), and MLi2 (250 nM) as indicated. (B, C) Relative (fold of non-treated cells) quantification of the ratio of pRab10 to total Rab10 is shown, where each dot represents immunoblot analysis of one biological replicate (n = 3 biologically independent experiments). Group means are shown. Error bars represent ± SEM. Significance was assessed by one-way ANOVA with Tukey's *post hoc* test, with **** representing P < 0.0001. (D–F) Representative immunocytochemistry of pRab10 (green), LAMP1 (blue), and tubulin (red) from BMDMs (WT-LRRK2 or G2019S-LRRK2 as indicated) treated with nigericin (2 h at 5 μM) as indicated, then paraformaldehyde-fixed and saponin-treated. Scale bar: 10 μm. Quantification of pRab10 intracellular signal intensity in BMDMs determined from high-resolution confocal-reconstructed images, where each dot represents the analysis of pRab10 signal intensity in one cell, with ≥20 cells analyzed each from three biological replicates. Group means are shown. Error bars represent ± SEM. Significance was assessed by one-way ANOVA with Tukey's post hoc test, with **** representing P < 0.0001. (G) Representative immunoblots of BMDMs lysates from WT-LRRK2 BMDMs treated with or without nocodazole (10 μM, 2 h) and supplemented (as indicated) with GTP (1 mM), and lysates analyzed by immunoblotting. Relative (fold of non-treated cells) quantification of the ratio of pRab10 to total Rab10 is shown, where each dot represents immunoblot analysis of one biological replicate (n = 3 biologically independent experiments). Group means are shown. Error bars represent ± SEM. Significance was assessed by one-way ANOVA with Tukey's *post hoc* test, with ** representing P < 0.01. Exact P value: P = 0.0032. (H) Representative immunocytochemistry of pRab10 (green) and tubulin (red) from BMDMs (WT-LRRK2) treated with nocodazole following GTP supplementation, as indicated. Each dot represents the analysis of pRab10 signal intensity in one cell, with ≥20 cells analyzed each from three biological replicates. Group means are shown. Error bars represent ± SEM. Significance was assessed by one-way ANOVA with Tukey's post hoc test, with **** representing P < 0.0001.

epitopes on LRRK2 that remain accessible in stabilized eGFP-LRRK2 protein bound to microtubules (Fernández et al, 2022). The UDD3 antibody does not yield appreciable immunocytochemistry signals in macrophages cultured from *Lrrk2* knockout mice (Fig. EV4A), suggesting specificity of the antibody in the approaches herein. Neither nocodazole, docetaxel, nor MLi2 changes the subcellular distribution of endogenous WT-LRRK2 or G2019S-LRRK2 protein in macrophages, with LRRK2 protein remaining distributed relatively evenly across the cytoplasm and largely excluded from the nucleus (Fig. EV4B,C).

## Nocodazole prevents LRRK2 activation with lysosomal stress in a GTP-dependent manner

Though LRRK2 has basal activity in maintaining pRab levels in macrophages, LRRK2 activity and Rab phosphorylation are known to markedly increase under stress conditions. Lysosomal stress triggers increased LRRK2 recruitment to lysosomal membranes in both mouse primary astrocytes and transfected HEK293T cells (Bonet-Ponce et al, 2020). Once localized to these membranes, LRRK2 can phosphorylate Rab substrates (Eguchi et al, 2018). Nigericin is a small molecule ionophore that activates LRRK2 and causes lysosomal stress and possible elevations of lysosomal pH through the rapid exchange of K$^+$ ions for H$^+$ ions across the lysosomal membrane (Dhekne et al, 2023). To examine whether nocodazole treatment and microtubule disruption could rescue nigericin-induced increases in LRRK2 activity and elevated pRab10 levels, we treated macrophages expressing WT-LRRK2 or G2019S-LRRK2 with nigericin in combination with docetaxel or nocodazole. Immunoblot analysis demonstrated the ratio of pRab10 to total Rab10 does not change with microtubule stabilization in docetaxel treatment, but reduces with nocodazole treatment in both WT-LRRK2 and G2019S-LRRK2 expressing cells (Fig. 2A–C). Immunocytochemical analysis demonstrates that under nigericin-treatment conditions, Rab10 intracellular signal in fixed and saponin-treated cells markedly reduces compared to vehicle-treated cells (Fig. 2D–F).

Nocodazole treatment promotes tubulin GTPase activity in macrophages to rapidly facilitate microtubule disassembly (Dimitrov et al, 2008). LRRK2, encoding the Rab-like ROC domain, may require GTP-binding for kinase activation and Rab phosphorylation (Taymans et al, 2011; Galicia et al, 2024; Liu et al, 2016). To test the role of cellular GTP on Rab localization and LRRK2 activity in the presence of nocodazole, 1 mM GTP was combined in the macrophage cell culture media with streptolysin O permeabilization to facilitate supplemented GTP internalization to the macrophage cytoplasm (Blanca Ramírez et al, 2017). This supplemented level of GTP, but not GDP supplementation, restored the ratio of pRab10 to Rab10 in the presence of nocodazole (Fig. 2G). The restoration of intracellular pRab10 signal was confirmed with immunocytochemistry (Fig. 2H). These results implicate a GTP-sensitive mechanism, potentially unrelated to LRRK2 interactions with microtubules, for nocodazole treatment effects on the suppression of LRRK2-mediated pRab10.

## LRRK2 recruitment to the lysosome in non-stressed cells, but not to microtubule surfaces, promotes LRRK2-mediated Rab phosphorylation

With discrepant results between nocodazole and docetaxel on LRRK2-mediated Rab phosphorylation, and the implication that nocodazole may exert an indirect GTP-dependent effect on LRRK2, we next sought a molecular trapping method to provide further clarity into possible mechanisms of microtubule-surface-mediated regulation of Rab phosphorylation. The trapping method that can stabilize LRRK2 on different surfaces in the cell is based on the rapamycin-inducible FKBP-FRB system (Nihongaki et al, 2021). Previous efforts to stabilize over-expressed LRRK2 protein onto microtubules required over-expression of LRRK2 complexed with type I kinase inhibitors (i.e., the MLi2 small molecule inhibitor), obviating the possibility of measuring phosphorylated Rabs. Here we cloned custom plasmids expressing FKBP-fused WT-LRRK2 and G2019S-LRRK2, and co-transfected these plasmids with either FRB-fused mCherry as a negative control for cytoplasmic proteins, LAMP1-FRB that is known induce LRRK2 Rab phosphorylation (Kluss et al, 2022a), and FRB-fused ß-tubulin and EMTB-FRB (microtubule-binding domain of ensconsin) to stabilize FKBP-tagged proteins onto outer microtubule surfaces (Appendix Fig. S1). Both FRB-tagged ß-tubulin and EMTB baits can position target proteins to the immediate outer surface of dynamic microtubules

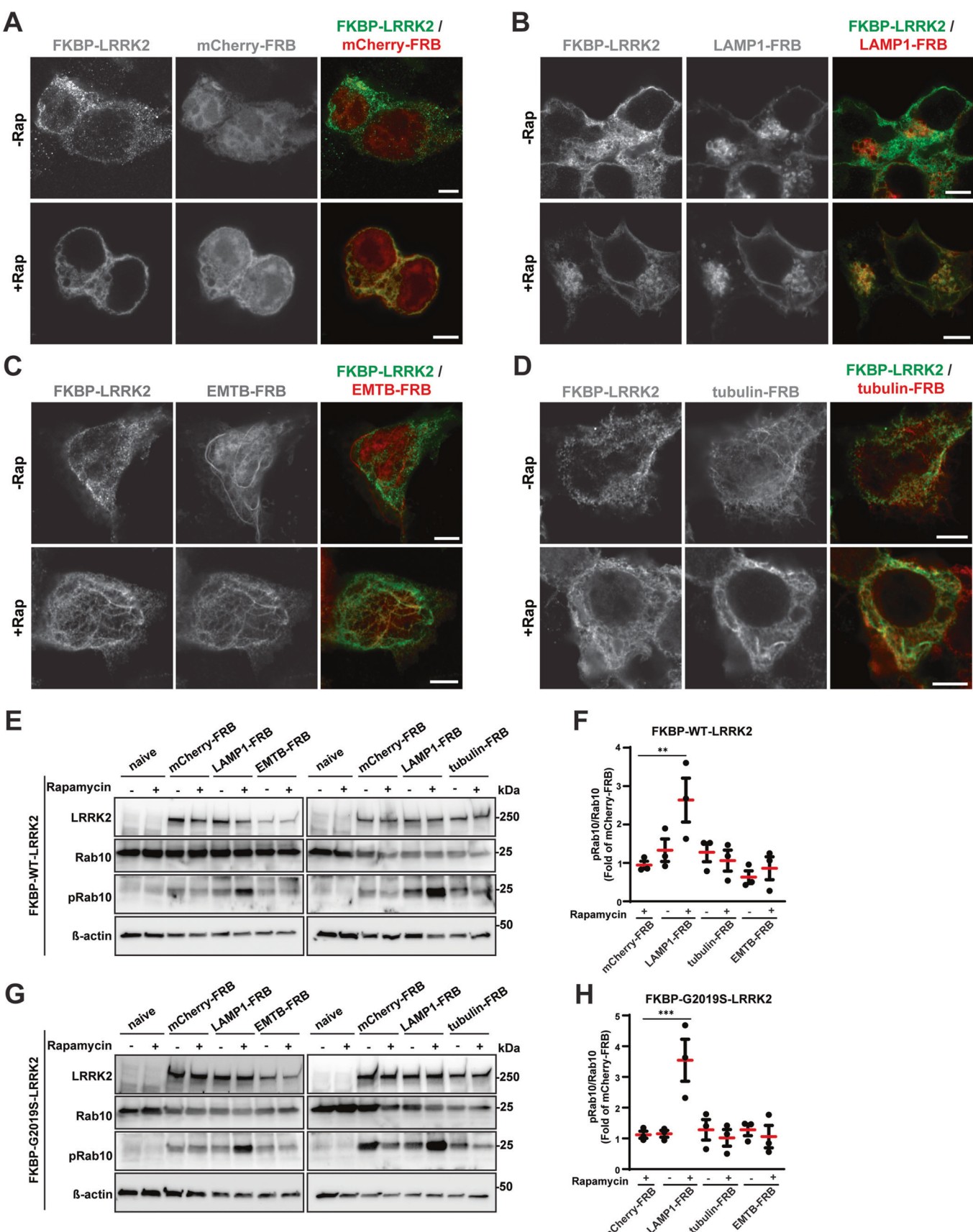

**Figure 3. LRRK2 recruitment to the lysosome, but not to microtubules, promotes LRRK2-mediated Rab phosphorylation.**

(A–D) Representative immunocytochemistry of transfected FKBP-WT-LRRK (green) HEK293T cells, co-transfected with mCherry-FRB (red), LAMP1-FRB (red), or tubulin-FRB (red), with or without rapamycin (100 nM, 30 min) treatment. Scale bars: 10 μm. Higher-magnification views of the same cells are shown in Appendix Fig. S1B. (E, F) Representative immunoblots of HEK293T cell lysates transfected with FKBP-WT-LRRK2 (except for lysates in 'naive' lanes) along with the indicated plasmid, and treated with or without rapamycin (100 nM, 30 min) prior to lysis. Relative (fold of FKBP-WT-LRRK2/mCherry-FRB transfected cells with rapamycin) quantification of the ratio of pRab10 to total Rab10 is shown, where each dot represents immunoblot analysis of one biological replicate ($n = 3$ biologically independent experiments). Group means are shown. Error bars represent ± SEM. Significance was assessed by one-way ANOVA with Tukey's post hoc test, with ** representing $P < 0.01$. Exact $P$ value: $P = 0.0089$. (G, H) The same experiments as in (E) and (F) but with HEK293T cells transfected with FKBP-G2019S-LRRK2 (except for lysates in 'naive' lanes; $n = 3$ biologically independent experiments). Significance was assessed by one-way ANOVA with Tukey's post hoc test, with *** representing $P < 0.001$. Exact $P$ value: $P = 0.0009$.

and have been used previously to understand the effects of microtubule interactions with other microtubule interacting proteins (Liu et al, 2022; Nihongaki et al, 2021).

HEK293T cells were transfected with combinations of FKBP-WT-LRRK2 and FKBP-G2019S-LRRK2 (N-terminal tags) with mCherry-FRB, LAMP1-FRB, EMTB-FRB, and tubulin-FRB (Fig. 3). Cells were treated with rapamycin to rapidly induce FRB-FKBP dimerization within minutes. Immunofluorescence staining patterns confirmed the lack of FKBP-fused proteins with FRB-fused proteins without the presence of rapamycin. Shortly after rapamycin treatment, FKBP-fused LRRK2 (WT or G2019S) robustly colocalized with all the FRB-tagged proteins in co-transfected cells (Fig. 3A–D). LRRK2 fusion to soluble mCherry did not obviously change typical LRRK2 distribution in the cell or pRab10 levels (Fig. 3A,E–H; Appendix Fig. S1). Despite prominent mCherry-FRB localization to the nucleus, LRRK2-positive complexes with mCherry-FRB did not form in the nucleus after rapamycin treatment. LRRK2 trapping to LAMP1-FRB induced a significant ~2.5-fold increase (WT-LRRK2) and ~3.5-fold increase (G2019S-LRRK2) in pRab10 levels (Fig. 3B,E–H) in non-activated and non-stressed macrophages. Although both ß-tubulin-FRB and EMTB-FRB targeting strongly induced LRRK2 relocalization to available microtubules in the cells (Fig. 3C,D; Appendix Fig. S1), there were no apparent changes in pRab10 levels (Fig. 3E–H). Together, these results suggest that the relocalization of LRRK2 to tubulin or microtubule surfaces has a negligible effect on pRab10 levels, at least in transfected HEK293T cells.

To explore these interactions further in macrophage cells, the murine Raw 264.7 cell line (ATCC SC-6004 *Lrrk2*$^{-/-}$) deficient in endogenous *Lrrk2* was transfected with FKBP-WT-LRRK2 or FKBP-G2019S-LRRK2 and LAMP1-FRB plasmids, and treated with or without nocodazole (Fig. 4). Due to the relatively poor transfection efficiency in the cells (e.g., 10–20%), immunoblotting did not have sensitivity to resolve pRab10 signal that is absent in untransfected cells. However, 2 h of nocodazole treatment led to a ~50% reduction of intracellular pRab10 signal according to the immunocytochemical analysis of cells expressing FKBP-LRRK2 and LAMP1-FRB proteins (Fig. 4A–D).

## LRRK2 interactions with tubulin are independent of polymerization and LRRK2 kinase activity

Previous reports have demonstrated that over-expressed LRRK2 can bind to microtubules in cells in an organized fashion, and this process is promoted by some pathogenic *LRRK2* mutations as well

as binding to type I inhibitors (Kett et al, 2011; Deniston et al, 2020; Snead et al, 2022; Watanabe et al, 2020). To better understand LRRK2 interactions with tubulin, we employed a fast proximity biotinylation proteomic technique using APEX2 (Tan et al, 2020) fused (N-terminal) LRRK2 construct. Proteins nearby LRRK2 that might include free tubulin (e.g., α, ß, or γ), or tubulins incorporated into microtubules, are labeled with biotin over the course of a 1 min peroxide exposure (Appendix Fig. S2). While peroxide and oxidative stress are known to activate LRRK2, we and others have not recorded effects of oxidative stress on LRRK2 or Rab phosphorylation in shorter term exposures (e.g., less than 2 h, see (Fernández et al, 2022)). In cells expressing APEX2-LRRK2, the type I inhibitor MLi2 decreases levels of pRab10 expression (Fig. 5A,B). With FLAG-APEX2-only expression as a control, cytoplasmic APEX2 expression and activity in total biotinylated proteins assessed by whole proteomic analysis were similar across experiments expressing WT-LRRK2 (with or without MLi2) and G2019S-LRRK2 (Fig. EV5). The proportion of tubulin biotinylated by APEX2-tagged LRRK2 protein in all three conditions (WT-LRRK2 ± MLi2 or G2019S-LRRK2) was increased between 5–15-fold over APEX2-only enzyme (Fig. 5C), suggesting a much higher probability of close proximity of LRRK2 protein to tubulin than the cytoplasmic APEX2 enzyme by itself. However, the total amount of labeled tubulin did not change with MLi2 treatment or with the G2019S mutation, showing LRRK2 proximity to tubulin is independent of kinase activity. LRRK2 proximity interactions with Rab substrates (Rab7L1, Rab8a, Rab10, Rab35) were also greatly enriched over APEX2-only controls, but were also not overtly different between kinase overactive G2019S-LRRK2 or with MLi2 kinase inhibitor treatment (Fig. 5C).

To further assess LRRK2 interactions with microtubules under different combinations of nocodazole, docetaxel, and MLi2 treatment, total biotinylated proteins labeled by the APEX2-LRRK2 were isolated via direct streptavidin pulldowns followed by tubulin measurements via targeted affinity blots with anti-ß-tubulin antibodies (Fig. 5D; Appendix Fig. S2). Quantification of tubulin in proximity to LRRK2 demonstrated that LRRK2 interacts with a similar proportion of tubulin whether or not tubulin is in a polymerized state, and this is independent of LRRK2 kinase activity (Fig. 5E,F). While LRRK2 preferentially and transiently interacts with all tubulin subunits, especially γ-tubulin, these interactions are independent of MLi2 binding and tubulin polymerization. Further, LRRK2 interactions with Rab protein substrates Rab7L1, Rab8a, Rab10, and Rab35 are also insensitive to whether LRRK2 protein is bound to MLi2 or not.

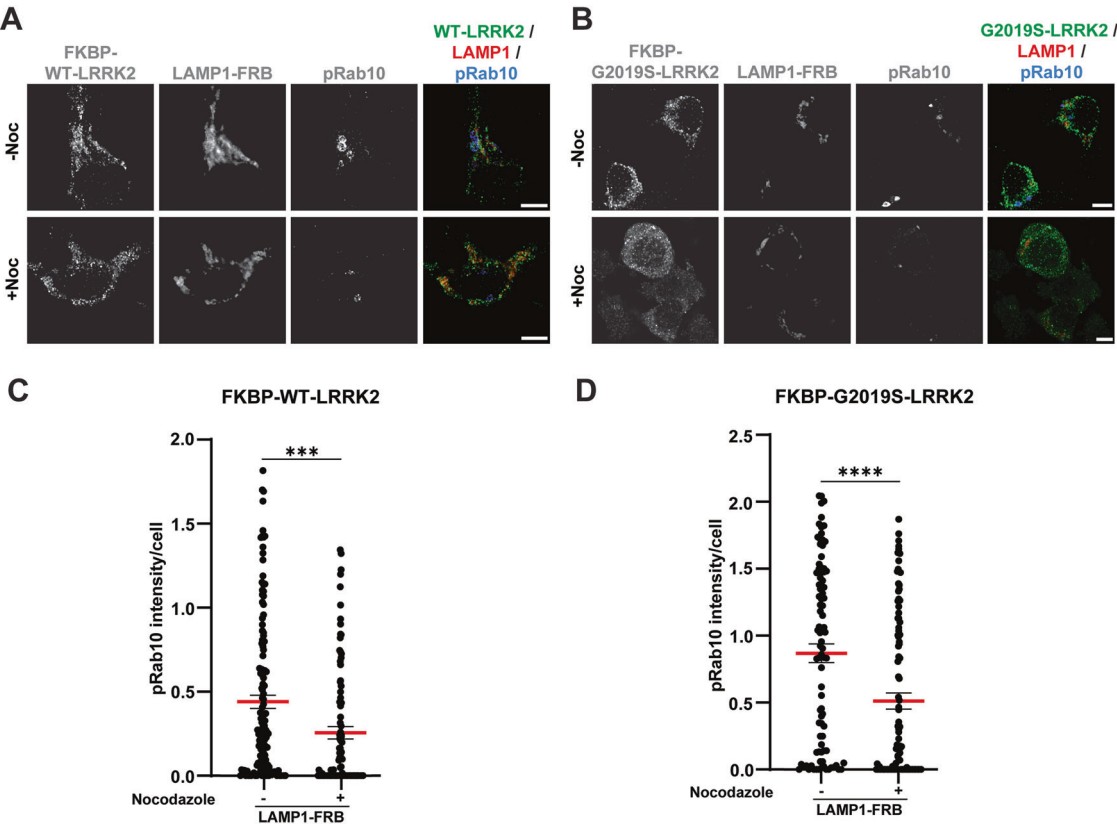

**Figure 4. Nocodazole attenuates Rab phosphorylation at the lysosome.**

(A, B) Representative immunocytochemistry of RAW-264-(*Lrrk2*$^{-/-}$) macrophages transfected with plasmids expressing FKBP-WT-LRRK2 or FKBP-G2019S-LRRK2 (green), LAMP1-FRB (red), and pRab10 (cyan), with and without nocodazole treatment (2 h at 10 µM). Rapamycin (100 nM) was included in all conditions 30 min prior to paraformaldehyde fixation and saponin treatment. Scale bars: 10 µm. (C, D) Quantification of relative pRab10 intracellular signal intensity in LRRK2-positive (i.e., transfected) macrophages determined from widefield fluorescent photomicrographs, where each dot represents the analysis of pRab10 signal intensity in one cell, with ≥20 cells analyzed each from three biological replicates. Group means are shown. Error bars represent ± SEM. Significance was assessed by one-way ANOVA with Tukey's post hoc test, with *** representing $P < 0.001$, **** representing $P < 0.0001$. Exact $P$ value for (C): $P = 0.0002$.

## Bulky N-terminal tags stabilize transfected LRRK2 protein onto microtubules

Several recent studies employed LRRK2 protein fused to N-terminal fluorescent proteins that were transiently expressed in cell lines to study LRRK2 interactions with microtubules (Snead et al, 2022; Fernández et al, 2022; Watanabe et al, 2020). Through this approach, type I inhibitors robustly stabilized an active (closed) LRRK2 conformation on microtubules (Snead et al, 2022). Given the discrepant results here, we hypothesized that the non-physiological high expression that closer approximates a 1:1 stoichiometry between tubulin and LRRK2 protein in combination with N-terminal bulky tags (e.g., eGFP or APEX2) might stabilize typically transient interactions between LRRK2 protein and microtubules. HEK293T cells were transfected with plasmids expressing LRRK2 with different N-terminal tags, FLAG-WT-LRRK2, eGFP-WT-LRRK2, mKate2-WT-LRRK2, and FLAG-APEX2-WT-LRRK2, treated with or without MLi2 (Fig. 6A). While LRRK2 robustly formed skein-like structures along microtubules with all the bulky tags (including APEX2), similar distributions were not identified with the FLAG-LRRK2 protein according to immunocytochemical analysis with the UDD3-

LRRK2 monoclonal antibody (Fig. 6B). The population of cells transfected with eGFP-WT-LRRK2, mKate2-WT-LRRK2, and FLAG-APEX2-WT-LRRK2 plasmids had about ~60, ~20, and ~40% of cells presenting with robust LRRK2-positive skein-like structures, and these tended to be cells with high degrees of polymerized tubulin. Noticeably, the FLAG tag on its own is small (~1 kDa) compared to the other tags (~26 kDa each), suggesting the adjacent N-terminal domain may be significantly stabilizing the resultant complex along microtubules. C-terminal tags were not evaluated here since any C-terminal modifications of LRRK2 are known to greatly impair LRRK2 kinase function (Reynolds et al, 2014).

To evaluate the potential for stable LRRK2 interactions with microtubules, especially in complex with type I kinase inhibitors in cells other than macrophages and transfected cell lines, we employed an oral dosing strategy with the type I brain-penetrant molecule MLi2. Using a protocol previously developed for the detection of WT-LRRK2 in the mouse brain (West et al, 2014), brain tissues were evaluated by immunohistochemistry in mice treated with or without MLi2 (Fig. 6C). LRRK2 subcellular distribution in medium spiny neurons in the dorsal striatum, or excitatory projection neurons in the motor cortex, was similar

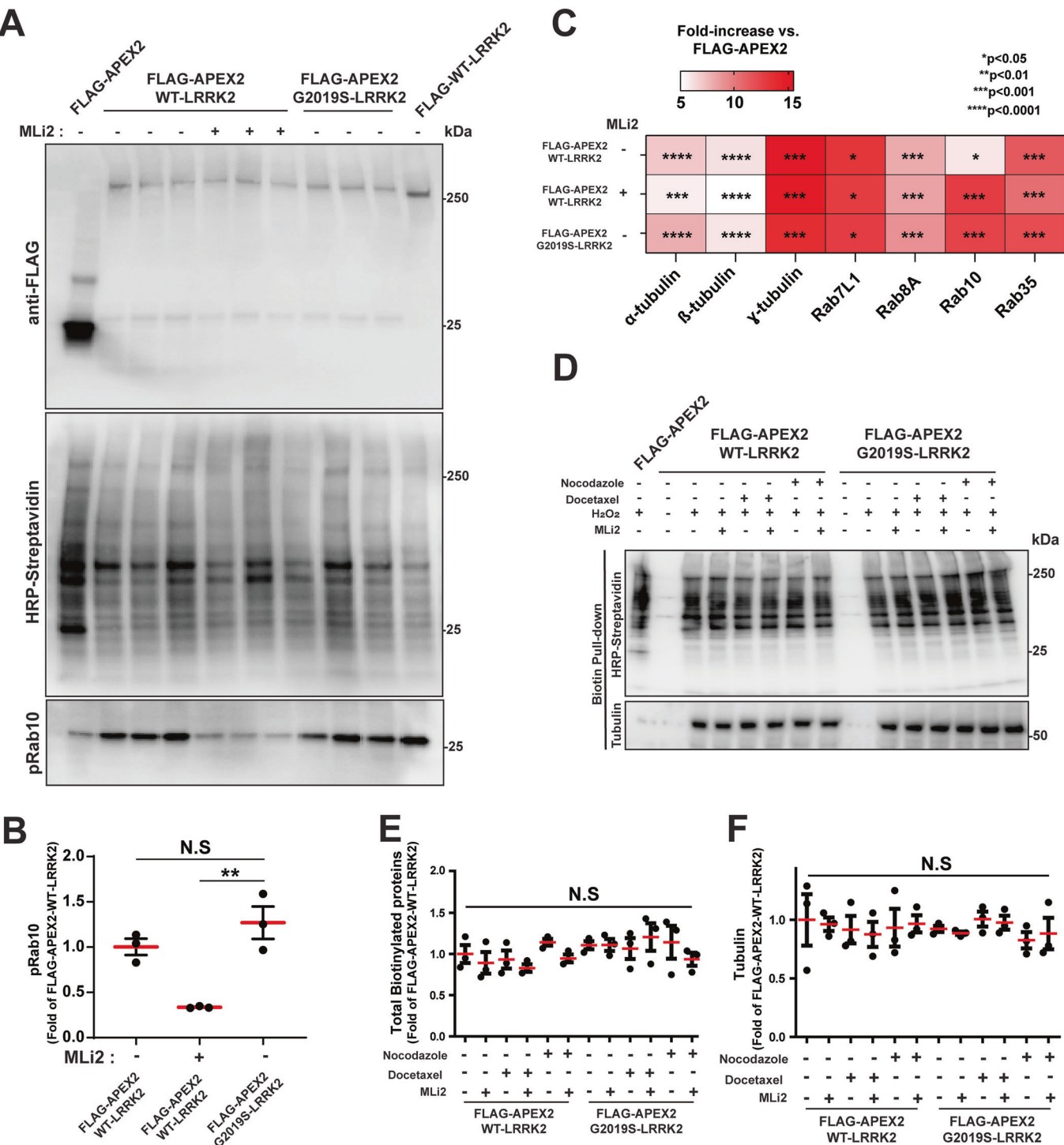

**Figure 5. LRRK2 interactions with tubulin are independent of microtubules and LRRK2 kinase activity.**

(A, B) Representative immunoblots of lysates from HEK293T cells transfected with FLAG-APEX2 (control), FLAG-APEX2-WT-LRRK2, FLAG-APEX2-G2019S-LRRK2, and FLAG-WT-LRRK2 (control) plasmids. Cells were treated with MLi2 (250 nM, 30 min) as indicated. Relative (fold of FLAG-APEX only) quantification of the ratio of pRab10 to total Rab10 measured by immunoblot, where each dot represents analysis of one biological replicate ($n = 3$ biologically independent experiments). Group means are shown. Error bars represent ± SEM. Significance was assessed by one-way ANOVA with Tukey's *post hoc* test, with ** representing $P < 0.01$. Exact $P$ value: $P = 0.0031$. (C) Quantification of seven selected biotinylated proteins from whole-cell proteomic measures of lysates from (A). (D–F) Representative immunoblots of biotin pull-downs from HEK293T lysates transfected with plasmids FLAG-APEX2 (control), FLAG-APEX2-WT-LRRK2, and FLAG-APEX2-G2019S-LRRK2. Prior to generating lysates for pull-downs, cells were treated with nocodazole or docetaxel (each 2 h at 10 μM), MLi2 (30 min at 250 nM as indicated), and $H_2O_2$ (1 min at 1 mM). Quantification of relative amount of total biotinylated proteins in the lysates in (E) and biotinylated tubulin (fold of FLAG-APEX-WT-LRRK2) in (F), where each dot represents the analysis of lysates from one of three biological replicates. Group means are shown. Error bars represent ± SEM. Significance was assessed by one-way ANOVA with Tukey's post hoc test.

compared to MLi2-naive mice (Fig. 6C). Colocalization between LRRK2 and the neuron-specific class III ß-tubulin (Tuj1) was also similar between MLi2-treated and naïve mice (Fig. 6D). The LRRK2 immunohistochemical signal was specific since *Lrrk2* knockout mice processed at the same time lacked signal for LRRK2. In evaluation of all regions of the brain known to harbor LRRK2 protein in neurons and other cells (West et al, 2014), we could not record a single instance of a LRRK2-positive skein in a cell. The effects of MLi2 in the mice on reducing pRab10 levels were confirmed using a biomarker platform measuring the ratio of pRab10 to total Rab10 (Appendix Fig. S3), consistent with our past reports of strong effects of MLi2 reducing brain pRab10 levels in these mouse strains (Yuan et al, 2024). Together, these results suggest that stable LRRK2 interactions with microtubules, even in the presence of type I inhibitors, cannot be verified under physiological conditions.

## Discussion

LRRK2 kinase activity is unambiguously linked to neurodegenerative diseases through pathogenic mutations that upregulate kinase activity and Rab phosphorylation. Pathways that regulate LRRK2 activity are of interest to better understand LRRK2 function in both health and disease. Several recent influential studies detailed novel LRRK2 conformations stabilized by microtubules (Deniston et al, 2020; Watanabe et al, 2020). In those studies, it was demonstrated that the type I inhibitor MLi2 was responsible for stabilizing a LRRK2 structure with enhanced affinity for microtubules, leaving open the possibility that microtubules, in general, may regulate LRRK2 kinase activity, even if the interactions between LRRK2 and microtubules are transient without type I inhibitors in the complex. The active LRRK2 complexes might be further promoted on microtubules by pathogenic LRRK2 mutations, providing further cause for additional biochemical exploration of this potential pathway for LRRK2 regulation. While several studies have demonstrated that trapping LRRK2 on lipid membranes upregulates LRRK2 kinase activity, whether microtubules also function in a similar manner, trapping LRRK2 to a surface that might contact Rab proteins, was not explored previously. Addressing these questions, this study does not find support for the hypothesis that microtubules stabilize active LRRK2 protein or promote Rab phosphorylation. Rather, stable LRRK2 interactions with microtubules appear largely relegated to certain over-expression paradigms in cell lines that may exploit a normally transient interaction with tubulin isoforms.

This study finds that microtubule destabilization via nocodazole treatment may impair Rab activation and LRRK2-mediated Rab phosphorylation in macrophages and other cell types, effects that can be recovered with supplemented GTP. Tubulin concentration varies between 10 and 20 µM in mammalian cells (Shida et al, 2010; Hiller and Weber, 1978). The average total GTP concentration in cells is 500 µM (Traut, 1994), whereas the total free GTP pool in cytoplasm is around 30 µM or less (Wolff et al, 2022), therefore close to estimated concentrations of free tubulin in microtubule-absent cells. Tubulin depolymerization by nocodazole leads to the rapid accumulation of free tubulin in the GDP-bound

state thereby limiting the production of GTP. Consequently, nocodazole may transiently reduce the cytoplasmic GTP pool, which may affect LRRK2 GTP-binding potentially required for LRRK2-mediated Rab phosphorylation. Ours and others past studies show that LRRK2 affinity for GTP is low (Liu et al, 2016; Liu and West, 2016) compared to Rab proteins (Dumas et al, 1999; Simon et al, 1996), suggesting GTP-supplementation experiments may rescue nocodazole effects through upregulating GTP-bound LRRK2 protein. Though methods to measure GTP-bound LRRK2 in living cells have not yet been described, future studies that directly measure cellular free GTP levels as they may direct Rab substrate phosphorylation may be of interest in exploring rapid stress-activation and maintenance of LRRK2 phosphorylation pathways.

Nocodazole impairs Rab phosphorylation even when LRRK2 is kinetically trapped to membranes, or LRRK2 activity is induced with lysosomal damage. These results may be consistent with a feed-forward hypothesis involving Rab activation and reciprocal LRRK2 kinase activation at the membrane that regulates the production of phosphorylated Rabs (Vides et al, 2022). LRRK2 interaction with microtubules is transient and largely independent of either LRRK2 kinase activity or tubulin polymerization status, and the stabilization of microtubules in cells has little to no effect on LRRK2-mediated Rab phosphorylation. Trapping FKBP-LRRK2 to soluble mCherry-FRB bait also had inconsequential effects on LRRK2-mediated Rab phosphorylation, further supporting the FKBP-FRB system as a robust one to dissect LRRK2 co-factor function. Notably, factors that regulate the initial recruitment of LRRK2 substrate Rab proteins to the membrane remain largely unknown, but the identification of such factors could provide critical clues as to the underlying physiological regulation of LRRK2 activity in both health and disease.

It is widely appreciated that centrosomes serve as the main microtubule-organizing centers in most cells and play a critical role in cell division, cilia formation, and cell polarity. Indeed, some studies suggest important interactions between LRRK2 and γ-tubulin (Lara Ordónez et al, 2019; Naaldijk et al, 2024). This study supports and adds to the understanding of LRRK2 interactions with γ-tubulin which likely occurs at the centrosome. The function of LRRK2 at the centrosome, and purpose of transient interactions with tubulins and microtubules remain to be clarified. Appreciating that LRRK2 is a potentially aggregation prone protein, transient interactions with abundant tubulins and microtubules may facilitate LRRK2 distribution across the cytoplasm, creating availability if and when the enzyme is needed in complex with Rab proteins at the membrane. While most experiments were performed in this study with non-stressed cells, additional experimentation, for example fast-scanning proximity experiments, in combination with different damaging agents (e.g., microtubule destabilizers or lysosomotropic agents) may provide new insights as to why LRRK2 may phosphorylate Rab proteins under stress conditions and how tubulin interactions may interact. However, based on the results in this study, caution may be warranted for continued reliance on over-expressed tagged LRRK2 protein. Replication of interactions under physiological conditions may be required before interpreting pathway function for LRRK2 regulation in health and disease.

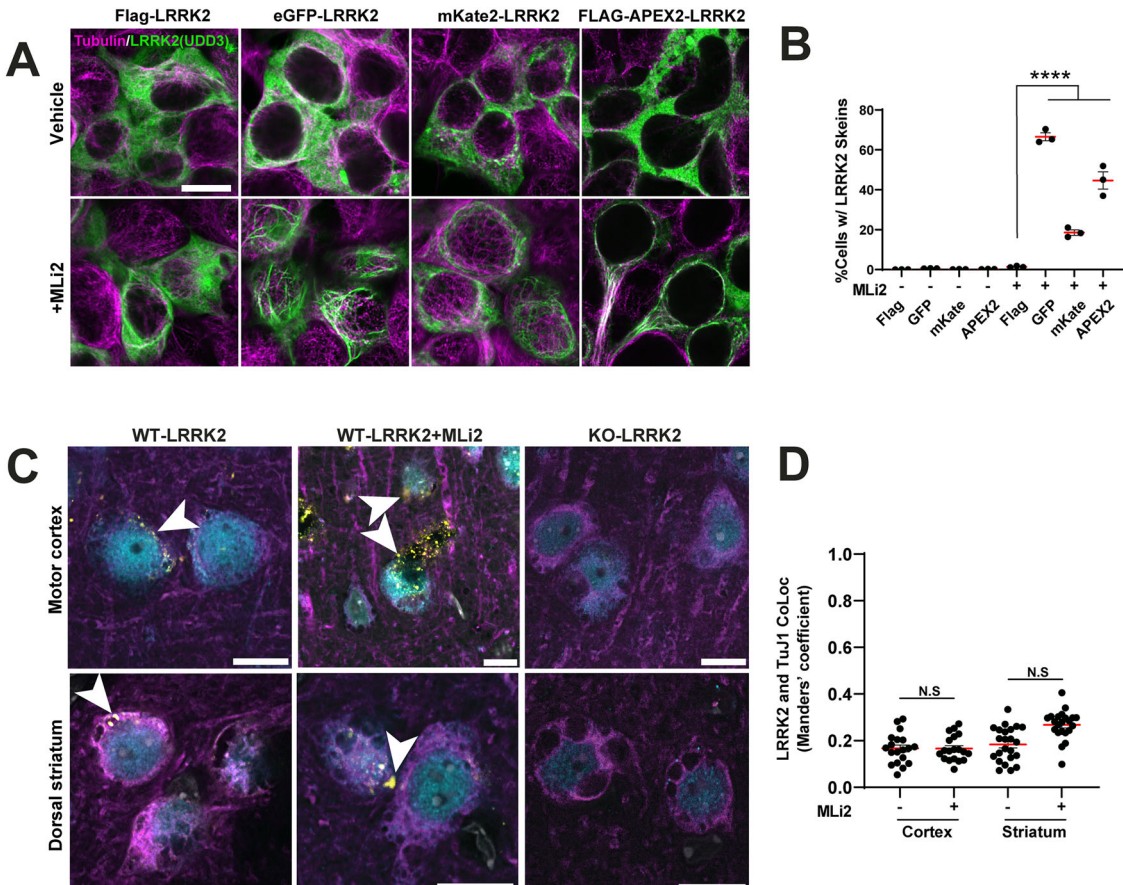

**Figure 6. Bulky N-terminal tags stabilize LRRK2 protein onto microtubules.**

(A, B) Representative immunocytochemistry of HEK293T cells transfected with plasmids for LRRK2 with different N-terminal tags (green, with the indicated N-terminal tag), and tubulin (magenta). Prior to paraformaldehyde fixation and immunostaining, cells were treated with vehicle (DMSO) or MLi2 (250 nM for 2 h) as indicated. Scale bars: 10 μm. Quantification of cells positive for LRRK2-positive skein-like structures, where each dot represents the mean of the analysis of ≥25 cells each from 3 biological replicates. Group means are shown. Error bars represent ± SEM. Significance was assessed by one-way ANOVA with Tukey's post hoc test, with **** representing $P < 0.0001$. (C, D) Representative immunohistochemistry for endogenous Lrrk2 (yellow), TuJ1 (magenta), and NeuN (cyan) immunostaining in WT-LRRK2 and $Lrrk2^{-/-}$ mouse brain tissues, either from motor cortex or dorsal striatum as indicated. White arrows indicate representative LRRK2-positive puncta resolved in NeuN-positive cell bodies. Scale bars: 10 μm. Quantification of LRRK2-positive pixels colocalized with TuJ1 pixels. Manders' coefficients are shown after automatic thresholding where each dot represents the mean analysis of cells within a single confocal image, where at least 18 images were analyzed per group (>50 NeuN-positive neurons evaluated per group), representing the analysis of 2 and 3 animals per treatment group naive and MLi2-treated, respectively. Group means are shown. Error bars represent ± SEM. Significance was assessed by Student's $t$-test.

# Methods

### Reagents and tools table

| Reagent/resource | Reference or source | Identifier or catalog number |
|---|---|---|
| **Experimental models** | | |
| Monocyte-derived macrophages | American Red Cross | NA |
| HEK293T | ATCC | CRL-3216 |
| B6;SJL-Tg(LRRK2)66Mjff/J (*M. musculus*) | The Jackson Laboratory | NA |
| C57BL/6J-Tg(LRRK2*G2019S)2AMjff/J (*M. musculus*) | The Jackson Laboratory | NA |

| Reagent/resource | Reference or source | Identifier or catalog number |
|---|---|---|
| B6.129×1(FVB)-Lrrk2tm1.1Cai/J (*M. musculus*) | The Jackson Laboratory | NA |
| RAW 264.7 | ATCC | TIB-71 |
| *Lrrk*2 KO RAW 264.7 | ATCC | SC-6004 |
| **Recombinant DNA** | | |
| 3xFLAG-FKBP-WT-LRRK2 | Addgene | #193584 |
| pCDNA3.1-3xFlag-WT-LRRK2 | (Smith et al, 2006) | NA |
| pCDNA3.1-3xFlag-G2019S-LRRK2 | (Smith et al, 2006) | NA |

| Reagent/resource | Reference or source | Identifier or catalog number |
|---|---|---|
| pCDNA3.1-FKBP-3xFlag-WT-LRRK2 | This study | NA |
| pCDNA3.1-FKBP-3xFlag-G2019S-LRRK2 | This study | NA |
| mKate2-WT-LRRK2 | Genscript | NA |
| pDEST53-LRRK2-WT (GFP-WT-LRRK2) | Addgene | #25044 |
| pCDNA3.1-APEX2-3xFlag-G2019S-LRRK2 | This study | NA |
| pCherry-FRB | Addgene | #25920 |
| Lamp1-mCherry-FRB | Addgene | #186576 |
| EMTB-CFP-FRB | (Liu et al, 2022) | NA |
| Tubulin-outer-FRB | (Nihongaki et al, 2021) | NA |
| **Antibodies** | | |
| Anti-LRRK2 (UDD3 30(12)) | Abcam | ab133518 |
| Anti-LRRK2 (N241A/34) | Antibodies Inc | 75-253 |
| Anti-β-tubulin | MP Biomedical | 691261 |
| Anti-α-tubulin-HRP | Abcam | ab40742 |
| Anti-β-actin | Santa-Cruz | sc-47778 HRP |
| Anti-Hsp70 | Thermo Fisher | PA5-28003 |
| Anti-LAMP1 | Santa Cruz | 1D4b |
| Anti-NeuN | GeneTes | GTX00837 |
| Anti-TuJ1 | Biolegend | 801202 |
| Anti-Rab10 (MJF-R23) | Abcam | ab237703 |
| Anti-Rab10 (phospho T73) (MJF-R21) | Abcam | ab230261 |
| Anti-Rab10 (phospho T73) (MJF-R21-22-5) | Abcam | ab241060 |
| Anti-Flag | Sigma | A9594 |
| Anti-V5 | Abcam | ab9116 |
| Anti-V5 | Bethyl | A190-119A |
| Anti-Flag-HRP | Sigma | A8592 |
| Streptavidin-TRITC | Jackson ImmunoResearch Inc | 016-020-084 |
| Streptavidin-HRP | Invitrogen | s-911 |
| **Chemicals, enzymes, and other reagents** | | |
| Fetal bovine serum | Atlanta Biological | S11150H (Q-39042) |
| DMEM | Gibco | 11995073 |
| Penicillin-streptomycin | Gibco | 15140122 |
| Antibiotic-antimycotic | Gibco | 15240062 |
| Mouse recombinant M-CSF | Biolegend | 576406 |
| Human recombinant M-CSF | Peprotech | AF-300-25-10UG |
| TrypLE Express | Gibco | 12605010 |
| Accutase | Stemcell Technologies Inc. | 07920 |
| Poly-D-lysine | Gibco | A3890401 |
| HBSS | Gibco | 14025092 |

| Reagent/resource | Reference or source | Identifier or catalog number |
|---|---|---|
| PEI | Polysciences | 24885 |
| Normal donkey serum | VWR | SD30 |
| Prolong Gold Antifade mountant | Thermo Fisher | P36934 |
| DTT | VWR | D11000-25 |
| Methanol | Fisher Scientific | A412-4 |
| DAPI | Thermo Fisher | D1306 |
| Hoechst | BD Bioscience | 561908 |
| Paraformaldehyde 32% | VWR | 15714-S |
| PBS endotoxin-free | VWR | 21040CM |
| Streptolysin O | Sigma | S5265 |
| Guanosine 5'-triphosphate sodium salt | Sigma | G8877 |
| Immobilon Classico Western HRP substrate | Millipore | WBLUC0500 |
| Immobilon Crescendo Western HRP substrate | Millipore | WBLUR0500 |
| Docetaxel | MCE | HY-B0011 |
| Nocodazole | MCE | HY-13520 |
| MLi2 | Pharmaron Inc | NA |
| Rapamycin | SelleckChem | s1039 |
| Biotinyl tyramide | Sigma | SML2135-250MG |
| Hydrogen peroxide | Sigma | HX0640-5 |
| Sodium azide | Sigma | S2002-5G |
| Sodium ascorbate | Sigma | 11140-250G |
| Trolox ((±)-6-Hydroxy-2,5,7,8-tetramethylchromane-2-carboxylic acid) | Sigma | 238813 |
| Tris base | RPI | T60040-5000.0 |
| Sodium chloride | Sigma | S7653-250G |
| Triton X100 | Sigma | T8787-250ML |
| PMSF | Thermo Fisher | AC2157400100 |
| cOmplete protease inhibitor cocktail | Roche | 11697498001 |
| PhosStop | Roche | 4906845001 |
| SDS | Sigma | L4509-1KG |
| Glycerol | RPI | G22020-4.0 |
| Bromophenol blue | Santa Cruz Biotechnology | sc-214633 |
| **Software** | | |
| GraphPad Prism v10 | http://www.graphpad.com | |
| FIJI | https://fiji.sc/ | |
| ImageLAB 6.1 | https://www.bio-rad.com/ru-ru/product/image-lab-software?ID=KRE6P5E8Z | |

| Reagent/resource | Reference or source | Identifier or catalog number |
|---|---|---|
| Zen Black 3.0 | https://www.micro-shop.zeiss.com/en/us/softwarefinder/software-categories/zen-black | |
| Olyvia | https://www.olympus-lifescience.com/en/downloads/detail-iframe/?0[downloads][id]=847252030 | |
| Adobe Photoshop | https://www.adobe.com/products/photoshop/ | |
| Adobe Illustrator | https://www.adobe.com/products/illustrator/ | |
| QuPath | https://qupath.github.io/ | |
| **Other** | | |
| Neon Transfection system 100 μL kit | Thermo Fisher | MPK10096 |
| EasySep Human Monocyte Enrichment Kit without CD16 Depletion | Stemcell Technologies Inc. | 19058 |
| Pierce BCA protein assay kit | Thermo Fisher | 23225 |

The Reagents and Tools Table provides comprehensive source and identifier or catalog numbers.

## Expression plasmids

A pCDNA3.1-3xFlag-LRRK2 construct previously described (Smith et al, 2006) was modified with site-directed mutagenesis to generate pCDNA3.1-3xFlag-G2019S-LRRK2. The FKBP tag sequence was amplified from an FKBP-pDEST plasmid kindly provided by Luis Bonet-Ponce and Mark Cookson and inserted in pCDNA3.1-3xFlag-LRRK2 with *NheI* and *KpnI* restriction sites. The N-term mKate2 and APEX2 tags were similarly inserted into LRRK2 plasmids. pCherry-FRB and LAMP1-mCherry-FRB were obtained from Addgene (Plasmids 25920, 186576). The EMTB-CFP-FRB plasmid was kindly provided by Dr. Yu-Chun Lin. The Tubulin-outer-FRB plasmid was kindly provided by Dr. Takanari Inoue.

## Animals

All the mice (males and females 3–5 months of age) used in this study were bred at Duke University with approval from the Institutional Animal Care and Use Committee. Mice originally developed in the laboratory of Zhenyu Yue and obtained from Jackson Laboratories include FLAG-WT-mLRRK2 BAC (B6.Cg-Tg(Lrrk2)6Yue/J), FLAG-G2019S-mLRRK2 BAC (B6.Cg-Tg(Lrrk2*G2019S)2Yue/J), and Lrrk2-KO (B6.129×1(FVB)-*Lrrk2tm1.1Cai*/J).

## Cell lines

Mouse bone marrow-derived macrophages (BMDMs) were generated by culturing the mouse bone marrow cells collected from 3- to 5-month-old mice into DMEM (Gibco) supplemented with 10% fetal bovine serum (Atlanta Biological), penicillin-streptomycin (Gibco), and mouse M-CSF (Biolegend). Human leukocytes were purchased from the American Red Cross, National Testing Lab-Charlotte, North Carolina. Samples were processed with the EasySep Human Monocyte Enrichment Kit without CD16 Depletion (Stemcell Technologies, Inc.). Purified monocytes were cultured in DMEM supplemented with Glutamax media (Invitrogen), 10% fetal bovine serum, Antibiotic-Antimycotic (Gibco), and human M-CSF (Peprotech). The cells were cultured for 7 days before the experiments to obtain human macrophage characteristics. *Lrrk2* knockout Raw 264.7 cells and HEK293T cells were cultured in DMEM supplemented with 10% fetal bovine serum and penicillin-streptomycin.

## Cell transfections and pharmacologic inhibition

Transfection of *Lrrk2* knockout RAW 264.7 cells was performed using a Neon transfection kit (Thermo Fisher) following the manufacturer's protocol. Transfection of HEK293T cells was performed using PEI (Polyscience) according to the manufacturer's protocol.

To stabilize and destabilize microtubules in BMDMs, human macrophages, HEK293T, and *Lrrk2* knockout RAW 264.7 cells were incubated for 2 h with 10 μM docetaxel and nocodazole, respectively. To induce lysosomal stress, cells were incubated for 2 h with 5 μM nigericin. To inhibit LRRK2 activity, cells were incubated for 2 h with 250 nM MLi2. To induce FRB and FKBP dimerization in transfected HEK293T and *Lrrk2* knockout RAW 264.7, cells were incubated for 30 min with 100 nM rapamycin.

## APEX2-mediated biotin labeling

APEX2-mediated biotin labeling was performed as previously described (Tan et al, 2020). Briefly, 48 h post-transfection, cells were incubated with 500 μM biotinyl tyramide (Sigma) for 30 min before labeling. Labeling was done by adding $H_2O_2$ 1 mM (Sigma) in PBS to the cell medium and swirled manually for 1 min. At 60 s, the media was poured out, and then the cells were washed with Quencher solution (Sodium Azide 10 mM, Sodium Ascorbate 10 mM, and Trolox 5 mM in PBS) three times. After washing, the cells were scraped and centrifuged at $15k \times g$ for 10 min at 4 °C. The cell pellets were lysed in RIPA buffer supplemented with Quencher solution (50 mM Tris-HCl pH 8.0, 150 mM NaCl, 1% TritonX-100, Sodium Ascorbate 10 mM, Sodium Azide 10 mM, Trolox 5 mM, PMSF 1 mM, Roche protease cOmplete Stop and phosphatase PhosStop inhibitor tablets). Zeba spin columns (Thermo Fisher) were used to remove the remaining free biotin phenol. Total protein concentration comparison was done using Pierce BCA Protein Assay Kit (Thermo Fisher) and stain-free SDS-PAGE precast gels (BioRad). Biotinylated protein pull-down was performed with Streptavidin magnetic beads (Thermo Fisher) according to the manufacturer's protocol. After the pull-down proteins were eluted using 2x Sample buffer (4% SDS, 20% glycerol, 120 mM Tris-HCl, bromophenol blue) supplemented with fresh 20 mM DTT and 2 mM free biotin. The samples were briefly vortexed and then placed in a thermo-mixer set to 70 °C, shaking at $1.2k \times g$ for 10 min. Beads were then pelleted on the magnetic rack and the protein eluate was collected.

## Immunoblotting

Protein lysates were analyzed using SDS–PAGE followed by transferring to PVDF membranes for immunoblotting with the

indicated primary antibodies and HRP-conjugated secondary antibodies. Signals were developed with Classico and/or Crescendo ECL reagent (Millipore) on a Chemidoc MP platform (BioRad). Saturated signals on immunoblots were not detected in any experiment used for analysis (ImageLab 6.1), and representative signals used for analysis are shown in the figures. The following antibodies were used for blotting: anti-LRRK2 (Antibodies Inc, 1:2000), anti-phospho-T73-Rab10 (MJF-R21, Abcam, 1:1000), anti-Rab10 antibody (MJF-R23, Abcam, 1:2000), anti-FLAG M2 (Sigma, 1:1000), Streptavidin-HRP (Thermo Fisher, 1:1000), HRP Anti-alpha Tubulin antibody (Abcam, 1:2000), anti-β-actin (Santa Cruz, 1:2000), anti-Hsp70 (Thermo Fisher, 1:2000).

## Mass spectrometry analysis

Mass spectrometry analysis was performed as previously described (Liu et al, 2020). Samples were spiked with undigested bovine casein at a total of either 1 or 2 pmol as an internal quality control standard. Samples were reduced with 10 mM DTT for 30 min at 80 °C and alkylated with 20 mM iodoacetamide for 30 min at room temperature. Then, samples were supplemented to a final concentration of 1.2% phosphoric acid with S-Trap (Protifi) binding buffer (90% MetOH/100 mM TEAB). Proteins were trapped, digested using 20 ng/μl sequencing grade trypsin (Promega) for 1 h at 37 °C, and eluted using 50 mM TEAB, followed by 0.2% formic acid, and lastly using 50% acetonitrile/0.2% formic acid. All samples were then lyophilized and resuspended in 12 μl 1% TFA/2% acetonitrile containing 12.5 fmol/μl yeast alcohol dehydrogenase (Sigma). Quantitative LC/MS/MS was performed using an MClass UPLC system (Waters Corp) coupled to a Thermo Orbitrap Fusion Lumos high-resolution accurate mass tandem mass spectrometer (Thermo Fisher) equipped with a FAIMSPro device via a nanoelectrospray ionization source. The sample was first trapped on a Symmetry C18 20 mm × 180 μm trapping column (5 μl/min at 99.9/0.1 v/v water/acetonitrile), after which the analytical separation was performed using a 1.8 μm Acquity HSS T3 C18 75 μm × 250 mm column (Waters Corp) with a 90-min linear gradient of 5–30% acetonitrile with 0.1% formic acid at a flow rate of 400 nl/min with a column temperature of 55 °C. Data collection on the Fusion Lumos mass spectrometer was performed in a data-dependent acquisition mode of r = 120,000 (@ $m/z$ 200) full MS scan from $m/z$ 375–1500 with a target AGC value of 4e5 ions. MS/MS scans were acquired in the ion trap in Rapid mode with an AGC target of 1e4 ions and a max injection time of 35 ms. The total cycle time for MS and MS/MS scans was 2 s. A 20 s dynamic exclusion was employed to increase the depth of coverage. The total analysis cycle time for each sample injection was ~2 h. Following UPLC-MS/MS analyses, data were imported into Proteome Discoverer 2.5 (Thermo Scientific Inc.). In addition to quantitative signal extraction, the MS/MS data was searched against the SwissProt H. sapiens database (downloaded in Nov 2019) a common contaminant/spiked protein database (bovine albumin, bovine casein, yeast ADH, etc.), and an equal number of reversed-sequence "decoys" for false discovery rate determination. Sequest (v 2.5, Thermo PD) with Infernys was utilized to produce fragment ion spectra and to perform database searches. Database search parameters included fixed modification on Cys (carbamidomethyl) and variable modification on Met (oxidation).

## GTP rescue assay

BMDMs were incubated with 10 μM nocodazole for 2 h and then treated with 20 ng/mL Streptolysin O in HBSS for 10–15 min at 37 °C. Next, the cells were incubated with Resealing buffer (10 mM HEPES, 140 mM NaCl, 5 mM KCl, 1.3 mM $MgCl_2$, 2 mM $CaCl_2$, pH 7.4) supplemented with 1 mM GTP for 10 min at 37 °C. Cells were lysed with 2x Laemmli buffer supplemented with 10% DTT followed by immunoblotting.

## Immunofluorescence

Cells were fixed with 4% paraformaldehyde, washed three times with PBS, permeabilized with 0.1% saponin, and blocked with 3% normal goat serum and immunostained with the following primary antibodies: anti-LRRK2 (UDD3, Abcam, 1:1000), phospho-T73-Rab10 (MJF-R21-22-5, Abcam, 1:1000), RAB10 antibody (Abcam, 1:1000), anti-FLAG-HRP (Sigma, 1:1000), anti-LAMP1 (Santa Cruz, 1:1000), anti-TuJ1 (Biolegend, 1:1000), anti-NeuN (GeneTes, 1:1000), Streptavidin-TRITC (Jackson ImmunoResearch, 1:2000), anti-V5 tag (Abcam, Bethyl, 1:1000). DAPI or Hoechst dyes were used for nuclear staining. Cell culture coverslips were mounted onto glass slides with ProLong Gold antifade reagent (Invitrogen). Microscopy images were obtained using Airyscan on a Zeiss 880 Inverted confocal microscope. Airyscan images and optimal Z-stacks were processed using Zen Black 3.0 software (Carl Zeiss).

## Microscopy

Microscopy images were obtained using Airyscan on a Zeiss 880 Inverted confocal microscope with a 63× oil immersion objective. Airyscan images and optimal Z-stacks were processed using Zen Black 3.0 software (Carl Zeiss). Transfected *Lrrk2* knockout RAW 264.7 cells and in BMDMs in GTP rescue assay cells were imaged using Olympus VS200 Research Slide Scanner with a 60× oil immersion objective. Images were processed using Olyvia and QuPath software.

## Image quantification

Images and fluorescence intensity profiles were analyzed using Fiji v2.16.0. Colocalization analysis was performed using Manders' coefficient after automatic thresholding in Fiji software.

## Statistical analysis

For primary cell culture experiments, an indicated number $n$ is the number of independent preparations (biological replicates) used for western blot analysis and immunofluorescence staining. In the case of colocalization studies, $n$ is the number of individual cells quantified in a given experiment, and every experiment was replicated at least three times on independent cell cultures. GraphPad Prism 10 was used to perform all statistical analyses: data distributions were estimated by Shapiro–Wilk tests. Significance was assessed by one-way ANOVA (with Tukey's group mean comparison post hoc test) or a two-tailed $t$-test for group comparisons. $P$ values < 0.05 were considered significant. The details of statistical analysis are specified in the figure legends. No blinding was performed.

## Data availability

The source data corresponding to each figure are deposited at Zenodo: https://doi.org/10.5281/zenodo.15206770.

The source data of this paper are collected in the following database record: biostudies:S-SCDT-10_1038-S44319-025-00486-6.

## Peer review information

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

## Acknowledgements

The authors sincerely thank Luis Bonet-Ponce and Mark Cookson for the FKBP-pDEST plasmid that was used for subcloning FKBP fragments into a common LRRK2 mammalian expression plasmid, as well as experimental advice regarding FRB constructs and stabilized microtubule binding conditions. The authors sincerely thank Dr.Yu-Chun Lin and Dr. Takanari Inoue for providing critical FRB constructs for this study. The authors thank Mike Lee and George Bloom for experimental advice regarding GTP supplementation and microtubule destabilization experiments. The authors thank Erik Soderblom and the rest of the Duke Proteomics Core Facility (http://www.genome.duke.edu/cores/proteomics/), which provided mass spectrometry proteomics services for this study. This study was supported by NIH R01-NS064934.

## Author contributions

**Tuyana Malankhanova**: Conceptualization; Resources; Data curation; Formal analysis; Supervision; Validation; Investigation; Visualization; Methodology; Writing—original draft; Project administration; Writing—review and editing. **Zhiyong Liu**: Conceptualization; Data curation; Formal analysis; Supervision; Investigation; Methodology; Project administration. **Enquan Xu**: Conceptualization; Data curation; Formal analysis; Investigation; Methodology. **Nicole Bryant**: Conceptualization; Data curation; Formal analysis; Methodology. **Ki Woon Sung**: Data curation; Investigation. **Huizhong Li**: Data curation; Formal analysis. **Samuel Strader**: Conceptualization; Data curation. **Andrew B West**: Conceptualization; Resources; Supervision; Funding acquisition; Visualization; Methodology; Writing—original draft; Project administration; Writing—review and editing.

Source data underlying figure panels in this paper may have individual authorship assigned. Where available, figure panel/source data authorship is listed in the following database record: biostudies:S-SCDT-10_1038-S44319-025-00486-6.

## Disclosure and competing interests statement

The authors declare no competing interests.

# Expanded View Figures

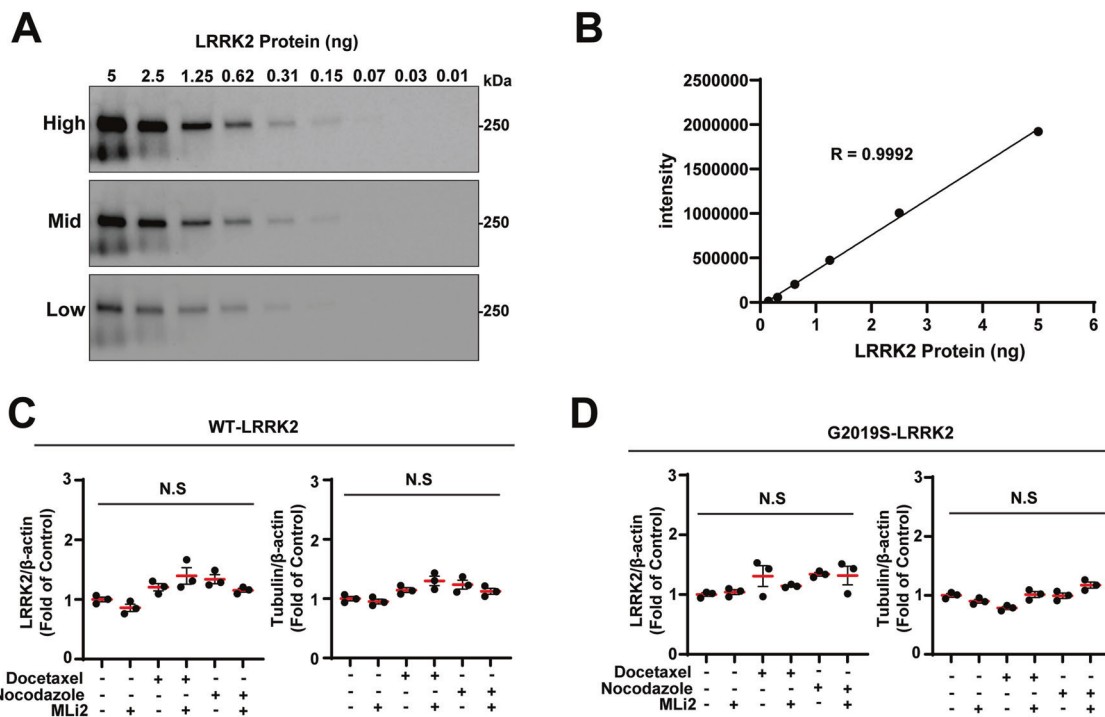

**Figure EV1.  LRRK2 levels in macrophages do not change with docetaxel, nocodazole, or MLi2 treatments.**

(A, B) Representative immunoblots of LRRK2 protein diluted two-fold to lower limits of detection, with a scatter plot highlighting good linearity (r = 0.99) of measured band intensities across the range of signals produced in the immunoblotting approaches used in this study. (C, D) Quantification of the relative (fold of non-treated cells) ratio of LRRK2 to ß-actin signals from immunoblots in Fig. 1A-B, where each dot represents immunoblot analysis of one biological replicate. Cells were treated with docetaxel (10 µM), nocodazole (10 µM), and MLi2 (250 nM) prior to lysis. Red bars in the column graphs show mean with ± SEM error bars. Statistical significance was assessed by one-way ANOVA; N.S, not significant.

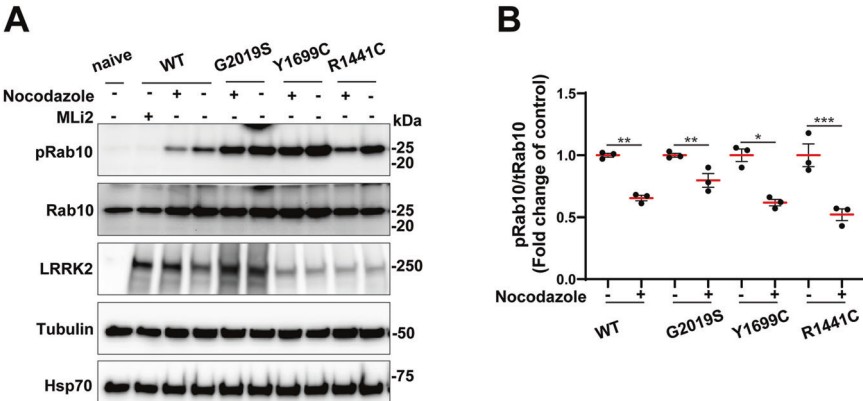

**Figure EV2. Nocodazole treatment partially inhibits transfected LRRK2 phosphorylation of endogenous Rab10 in HEK293T cells.**

(A, B) Representative immunoblots of HEK293T cell lysates transfected with the indicated LRRK2-expressing plasmid, treated with or without nocodazole (10 μM) and MLi2 (250 nM) for 2 h prior to lysis. Relative (fold of vehicle control) transfected cells for the quantification of the ratio of pRab10 to total Rab10, where each dot represents immunoblot analysis of one biological replicate ($n = 3$ biologically independent experiments). Group means are shown. Error bars represent ± SEM. Significance was assessed by Student's *t*-test with * representing $P < 0.05$, ** representing $P < 0.01$, *** representing $P < 0.001$. Exact $P$ values from left to right: $P = 0.0093$; $P = 0.0025$; $P = 0.0242$; $P = 0.0002$.

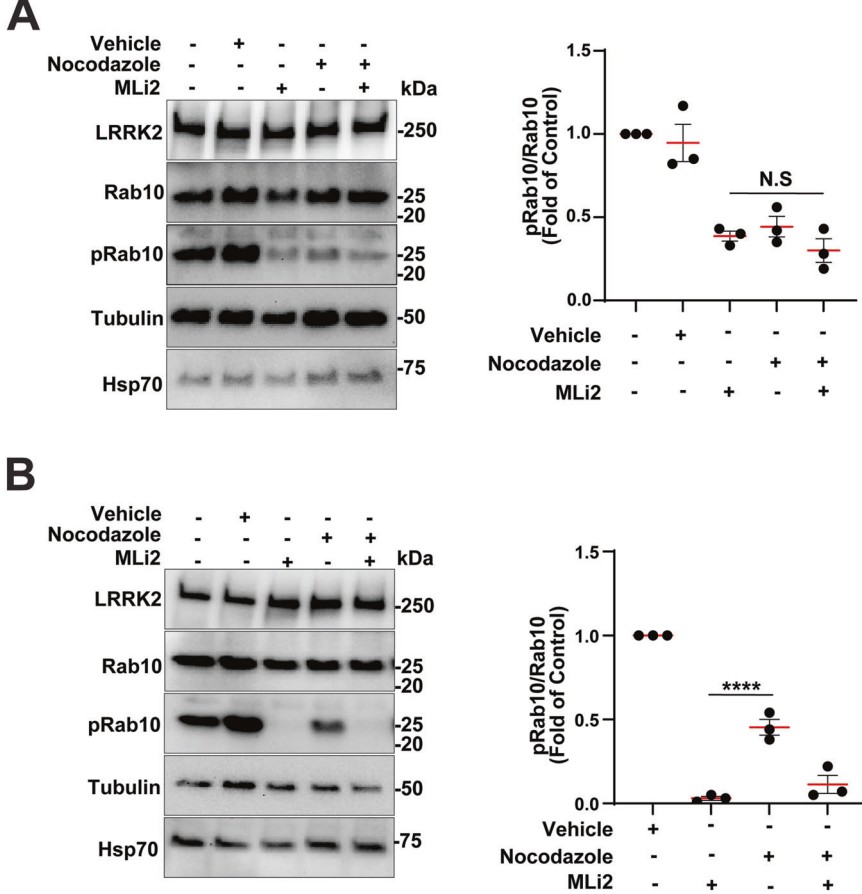

**Figure EV3. Nocodazole treatment does not hasten reductions of pRab10 after LRRK2 kinase inhibition.**

(A) Representative immunoblots of WT-LRRK2 BMDM cell lysates. Cells were treated with nocodazole (10 μM) and MLi2 (250 nM) for 2 min prior to lysis. Relative (fold of non-treated cells) quantification of the ratio of pRab10 to total Rab10 is shown, where each dot represents immunoblot analysis of one biological replicate ($n = 3$ biologically independent experiments). Group means are shown. Error bars represent ± SEM. Significance was assessed by one-way ANOVA with Tukey's *post hoc* test; N.S, not significant. (B) The same experiment as described in (A), but with a 5 min exposure of drugs (nocodazole and/or MLi2 as indicated), there is a partial loss of pRab10 caused by nocodazole treatment. Each dot represents one biological replicate ($n = 3$ biologically independent experiments). Group means are shown. Error bars represent ± SEM. Significance was assessed by one-way ANOVA with Tukey's *post hoc* test with **** representing $P < 0.0001$.

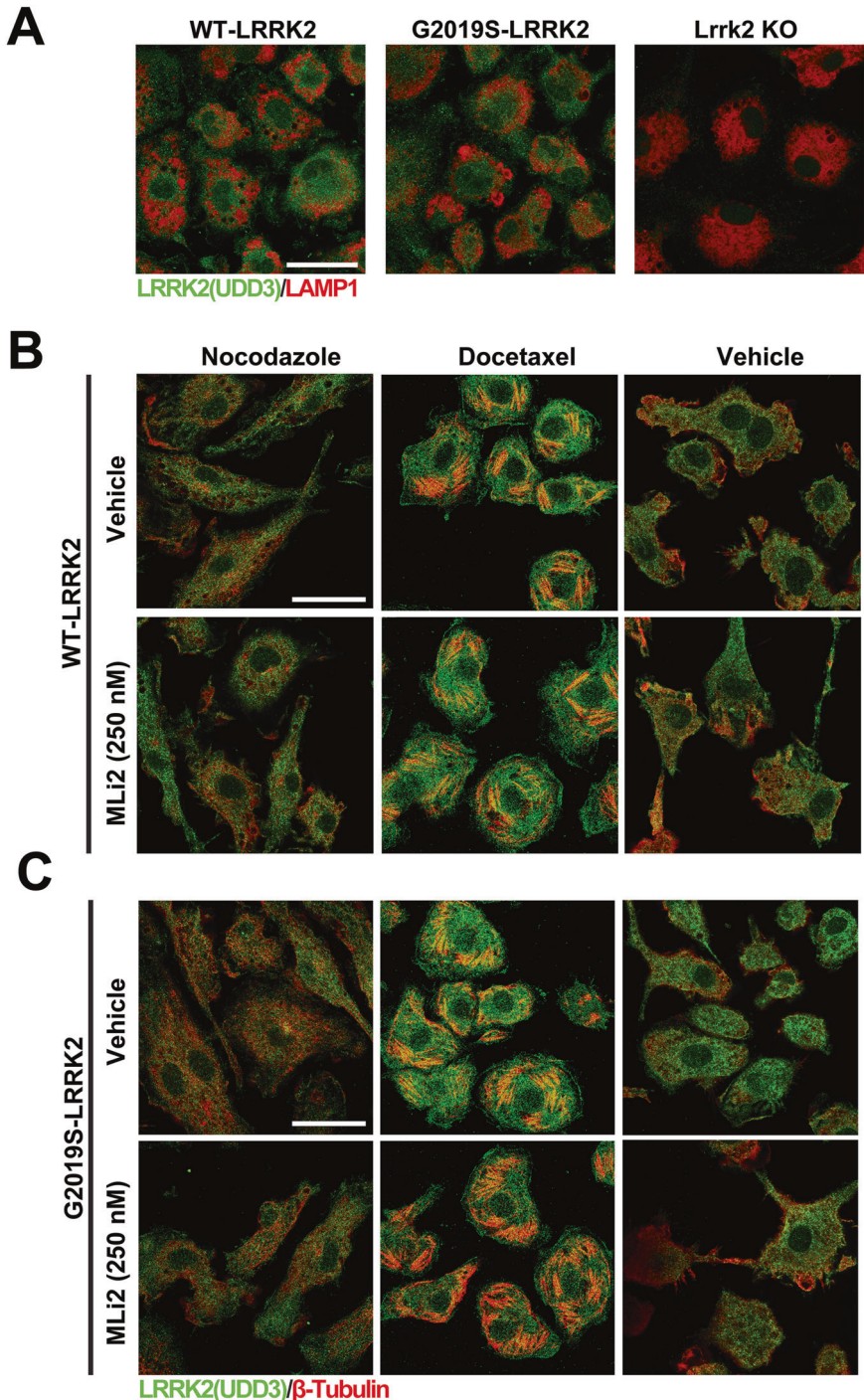

**Figure EV4. Nocodazole or docetaxel treatment does not change endogenous LRRK2 protein distribution in macrophages.**

(A) Representative immunocytochemistry of LRRK2 stained with the N-terminal targeting UDD3 antibody (green) and LAMP1 (red) from BMDM procured from WT-LRRK2, G2019S-LRRK2, or *Lrrk2* knockout mice. Cells were treated with docetaxel or nocodazole (10 µM, 2 h), then paraformaldehyde-fixed and saponin-treated prior to staining. There was no detectable LRRK2 signal from UDD3 in *Lrrk2* knockout macrophages, indicating specificity of the antibody in this immunocytochemistry protocol. Scale bars indicate 10 µm. (B, C) Representative immunocytochemistry showing endogenous WT-LRRK2 or G2019S-LRRK2 distribution is not affected by MLi2 (250 nM for 2 h) with or without docetaxel (10 µM) or nocodazole (10 µM), also treated for 2 h. No instances of LRRK2-skein-like structures or large (e.g., >1 µM) aggregates were noted in any cell in the experiments. Scale bars indicate 10 µm.

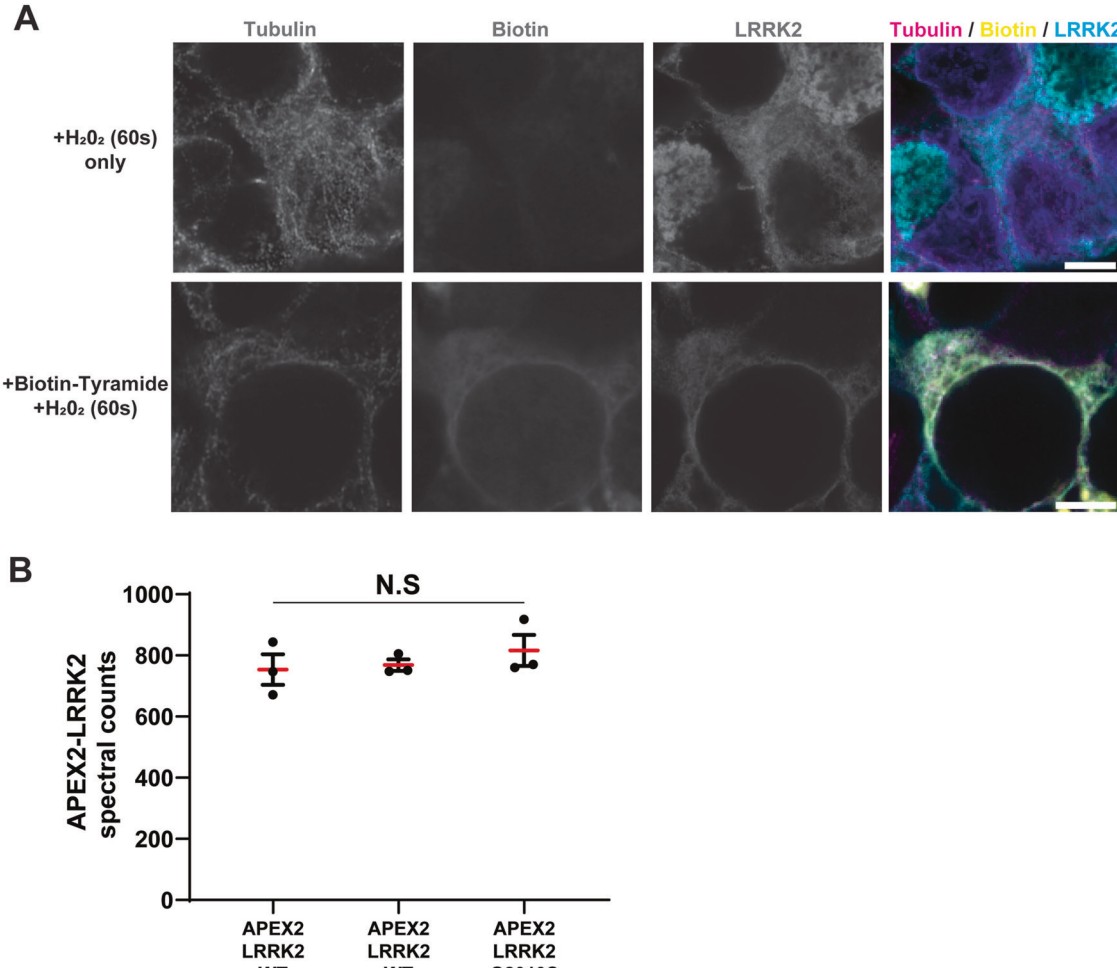

**Figure EV5. Subcellular localization and efficiency of APEX2-LRRK2 proximity labeled proteins.**

(A) Representative immunocytochemistry of transfected HEK293T cells with FLAG-APEX2-WT-LRRK2 plasmid, with and without biotin-tyramide and $H_2O_2$ supplementation. $H_2O_2$ treatment on its own did not result in measurable biotinylated proteins, whereas biotinylated proteins were measured when biotin-tyramide was included. Scale bars: 10 µm. (B) The overall concentration of biotinylated proteins captured with pull-downs from 3 biological replicates did not vary according to the presence of the G2019S-LRRK2 mutation or the LRRK2 inhibitor MLi2. Red bars in the column graphs show group means with ± SEM error bars. Statistical significance was assessed by one-way ANOVA; N.S, not significant.

