## [Peer Review File · EMBO Reports]

LRRK2 interactions with microtubules are independent of LRRK2-mediated Rab phosphorylation

Tuyana Malankhanova, Zhiyong Liu, Enquan Xu, Nicole Bryant, Ki Woon Sung, Huizhong Li, Samuel Strader, and Andrew West

Corresponding author(s): Andrew West (andrew.west@duke.edu)

Review Timeline:

Submission Date:	7th Aug 24
Editorial Decision:	9th Oct 24
Revision Received:	30th Dec 24
Editorial Decision:	18th Feb 25
Revision Received:	25th Apr 25
Accepted:	9th May 25

Editor: Deniz Senyilmaz Tiebe

Transaction Report:

Dear Prof. West,

Thank you for submitting your research manuscript to our journal, which was now seen by three referees, whose reports are copied below.

I apologize for this unusual delay in getting back to you, it took longer than anticipated to receive the referee reports. Also, three referees had initially agreed to review your manuscript. However, referee #2 has not returned his/her report despite our reminders.

Referees expressed in the proposed regulation of LRRK2 mediated Rab phosphorylation by microtubules. However, they, especially referee #1, also raised concerns that need to be addressed for publication here. In particular, referee #1 deems further insight into the proposed effect of nocadazole on Rab phosphorylation necessary - i.e. whether nocadazole activates Rab PPM1H phosphatase or whether nocadazole affects the localization of Rab12-activating vesicles need to be dissected. Moreover, referee #1 also finds that the quantification methods of Rab10 containing vesicles need to be revised. Please contact me if you have questions or comments regarding the revision and discuss any of the referee points further (also by video chat).

Should you be able to address the referee concerns satisfactorily, we would like to invite you to submit a revised manuscript. Please revise your manuscript with the understanding that the referee concerns (as in their reports) must be fully addressed and their suggestions taken on board. Please address all referee concerns in a complete point-by-point response. Acceptance of the manuscript will depend on a positive outcome of a second round of review. It is EMBO reports policy to allow a single round of major experimental revision only and acceptance or rejection of the manuscript will therefore depend on the completeness of your responses included in the next, final version of the manuscript.

We realize that it is difficult to revise to a specific deadline. In the interest of protecting the conceptual advance provided by the work, we recommend a revision within 3 months. Please discuss the revision progress ahead of this time with me if you require more time to complete the revisions, or if you have questions or comments regarding the revision (also by video chat).

1. A data availability section providing access to data deposited in public databases is missing (where applicable).
2. Your manuscript contains statistics and error bars based on $n=2$. Please use scatter plots in these cases.

You can submit the revision either as a Scientific Report or as a Research Article. For Scientific Reports, the revised manuscript can contain up to 5 main figures and 5 Expanded View figures, and it should not exceed 27000 characters. If the revision leads to a manuscript with more than 5 main figures it will be published as a Research Article. In this case the Results and Discussion section should be separate. If a Scientific Report is submitted, these sections have to be combined. This will help to shorten the manuscript text by eliminating some redundancy that is inevitable when discussing the same experiments twice. In either case, all materials and methods should be included in the main manuscript file.

4) a .docx formatted letter INCLUDING the reviewers' reports and your detailed point-by-point responses to their comments. As part of the EMBO publication's Transparent Editorial Process, EMBO reports publishes online a Review Process File (RPF) to accompany accepted manuscripts. This File will be published in conjunction with your paper and will include the referee reports, your point-by-point response and all pertinent correspondence relating to the manuscript.

<https://www.embopress.org/page/journal/14693178/authorguide#transparentprocess>

5) a complete author checklist, which you can download from our author guidelines

<https://www.embopress.org/page/journal/14693178/authorguide>. Please insert information in the checklist that is also reflected in the manuscript. The completed author checklist will also be part of the RPF.

6) Please note that all corresponding authors are required to supply an ORCID ID for their name upon submission of a revised manuscript (<<https://orcid.org/>>). Please find instructions on how to link your ORCID ID to your account in our manuscript tracking system in our Author guidelines

<<https://www.embopress.org/page/journal/14693178/authorguide#authorshipguidelines>>

7) Before submitting your revision, primary datasets produced in this study need to be deposited in an appropriate public database (see <https://www.embopress.org/page/journal/14693178/authorguide#datadeposition>). Please remember to provide a reviewer password if the datasets are not yet public. The accession numbers and database should be listed in a formal "Data Availability" section placed after Materials & Method (see also

<https://www.embopress.org/page/journal/14693178/authorguide#datadeposition>). Please note that the Data Availability Section is restricted to new primary data that are part of this study. * Note - All links should resolve to a page where the data can be accessed. *

Additional information on source data and instruction on how to label the files are available:

<https://www.embopress.org/page/journal/14693178/authorguide#sourcedata>

9) Our journal encourages inclusion of *data citations in the reference list* to directly cite datasets that were re-used and obtained from public databases. Data citations in the article text are distinct from normal bibliographical citations and should directly link to the database records from which the data can be accessed. In the main text, data citations are formatted as follows: "Data ref: Smith et al, 2001" or "Data ref: NCBI Sequence Read Archive PRJNA342805, 2017". In the Reference list, data citations must be labeled with "[DATASET]". A data reference must provide the database name, accession number/identifiers and a resolvable link to the landing page from which the data can be accessed at the end of the reference. Further instructions are available at <http://www.embopress.org/page/journal/14693178/authorguide#referencesformat>

10) Regarding data quantification (see Figure Legends:

<https://www.embopress.org/page/journal/14693178/authorguide#figureformat>)

12) Please also note our reference format:

13) All Materials and Methods need to be described in the main text using our 'Structured Methods' format, which is required for all research articles. According to this format, the Methods section includes a Reagents and Tools Table (listing key reagents, experimental models, software and relevant equipment and including their sources and relevant identifiers) followed by a Methods and Protocols section describing the methods using a step-by-step protocol format. The aim is to facilitate adoption of the methodologies across labs. More information on how to adhere to this format as well as a downloadable template (.docx) for the Reagents and Tools Table can be found in our author guidelines:

I look forward to seeing a revised version of your manuscript when it is ready. Please let me know if you have questions or comments regarding the revision.

Kind regards,

Deniz Senyilmaz Tiebe

Deniz Senyilmaz Tiebe, PhD
Senior Scientific Editor
EMBO Reports

Referee #1:

This generally, well written manuscript reports the surprising observation that LRRK2 phosphorylation of Rab10, Rab8 and Rab35 is strongly decreased in nocodazole-treated bone marrow derived macrophages, a condition in which they contain depolymerized microtubules. Nocodazole also interferes with pRab10 generation upon lysosome stress in these cells. Using streptolysin to add GTP to cells, the authors see less phosphorylation but it is not clear if this is important for LRRK2, Rab substrates or both. Locking LRRK2 on lysosomes enhances Rab phosphorylation, likely because the Rab is mistargeted there and cannot be retrieved by GDI; similar results have been reported by others upon organelle mistargeting in other cell types (Gomez). The authors conclude that nocodazole interferes with this lysosome activation but they may be using an invalid metric to conclude this. They confirm known protein proximities using APEX and conclude that bulky tags on LRRK2 increase the chances of inhibitor-driven aggregation that is non-physiological.

The story would be of great interest to readers of EMBO Reports if the authors probed the mechanism(s) of their unexpected nocodazole findings. There are several possible explanations. One is that nocodazole activates the Rab PPM1H phosphatase (known to prefer highly curved membranes that would be generated by Golgi fragmentation upon nocodazole addition). To test if nocodazole activates the Rab phosphatase, the authors should monitor the rate of dephosphorylation of phosphoRab10 from time zero of nocodazole addition by adding MLI-2 at time zero and tracking rate of phosphoRab10 decrease for various times. Another possibility is that nocodazole moves Rab12-activating vesicles away from those harboring Rabs 8, 10 or 35, leading to less LRRK2 activation. In this latter scenario, MLI-2 sensitive Rab12 phosphorylation would not change and this can easily (and should) be tested.

A major limitation of some of the analysis herein is a metric that the authors use of a "Rab10 vesicle" [Fig. 1G,H, 2 B,D,I, 4C,D: "pRab10 vesicles per cell"] At the EM level (and higher resolution light microscope level) there are surely more than 1.5 pRab10 vesicles-these cells contain ~2.5 million copies of Rab10, about half of which are membrane bound. In any case, it would be more correct to determine the total intensity of anti-pRab10 staining since the number of pRab10 structures don't serve as a useful indicator of total kinase activity and the value is not meaningful in relation to actual vesicle numbers. Again, 50 vesicles of a size below the diffraction limit, concentrated at the centrosome would look like one or two larger structures-and are harder to detect once dispersed by nocodazole. These graphs must all be corrected.

Additional comments:

The intro is written in a very scholarly manner

Figure 1. The blots in B and C are analyzed using non-linear ECL which will exaggerate and saturate signals. For example, Rab10 is highly overexposed and there is much less in lanes 5-8 in the wild type example. Lighter exposures of tubulin and Rab10 need to be shown or better yet, use of a fluorescent second antibody. In any case, nocodazole has a clear impact. (panels G and H are invalid). The authors state, "...there was an obvious reduction in Rab vesicle localization after nocodazole"...."stabilized in a GDI-bound cytoplasmic form" This has not been shown. To make this statement, the authors need to carry out membrane-cytosol cell fractionation. Dispersal of small (100K G X 60 minute sedimentable) vesicles is quite different than a change in Rab protein membrane association.

Figure 2. B, D, I are invalid; not sure what we learn from GTP addition experiment.

Figure 3. Please label the tags in relation to their N- or C-terminal locations in the constructs: It should be LAMP1-FRB for example. Does nocodazole have any effect in cells other than macrophages? Others seem not to see the nocodazole effect in LRRK2 R1441 MEFs (see Figure 6A, DOI: 10.1126/sciadv.adj1205). If macrophage-specific, please state this. Please indicate in legend where reader can find blots for panel E.

Figure 4. Constructs in this figure should be labeled "FKBP-". Images need to be re-quantified and #vesicles per cell removed (panels C and D).

Figure 5. As the authors note, 90% of LRRK2 is cytosolic. That tubulin is labeled by APEX is not a surprise since tubulin represents a huge fraction of total cellular proteins. Proximity to gamma tubulin confirms centriolar proximity, and APEX labeling of Rab29 (7L1), 10 and 35 are expected. Not sure what more one can conclude from this figure, despite the clean data. Since Rab8A, 10 and 35 bind to the LRRK2 ARM domain, insensitivity to MLI-2 is also not unexpected. Microscopy from many labs shows aggregates of overexpressed, inhibited LRRK2 along microtubules-but the reviewer agrees that this is non-physiological.

Figure 6. Here the authors conclude that large tags increase apparent microtubule-associated MLI-2 induced aggregation not seen in brain. Not sure how much one can conclude from changes in serum levels of LRRK2 {plus minus}MLI-2 (panel D)? The brain images show one neuron with LRRK2 signal. Detection of LRRK2 in brain is important (here also nicely controlled) and authors should quantify staining over 100 cells of a particular type to provide information related to LRRK2 distribution and distribution of cellular intensities. Really not sure why this data is included in this paper.

Referee #3:

This manuscript explores the consequences of microtubule stabilization and fragmentation on LRRK2 localization and activity. The work employs a rigorous application of modest exogenously expressed and endogenous LRRK2, and in some cases in the context of both WT and G2019S LRRK2. They find that microtubule depolymerization inhibits LRRK2 activation but the opposite is not true for microtubule stabilization. They importantly re-address reports of LRRK2 inhibitor induced association of LRRK2 protein with microtubules and data suggest these were largely artifacts of overexpression and tagging of LRRK2. While highly focused to interests of the LRRK2 community, the data are of high significance to that field given recent data provoking interest in LRRK2/microtubule interactions. There are only minor concerns for clarification.

1. For Fig 1 There is really no unaffected loading control protein for the western blots as LRRK2, Rab10, and tubulin are all affected in some way by the various treatments. Actin may be unwise later on as cytoskeleton dynamics are purposefully manipulated.
2. G2019S is a weak Rab kinase mutation - have the investigators looked at R1441C or Y1699C? These also differ in 14-3-3 binding.
3. The Docetaxel images in 1E do not reflect those in 1A. Clearer effects are seen in 1E.
4. There is a focus on Rab10 and pRab10 in the context of basal LRRK2 activity, lysosomal injury and membrane recruitment. Does one see the same effects with pRab8a (despite antibody specificity limitations) as it is likewise implicated in inflammatory responses and lysosomal injury? Can pS1292 be applied here for high endogenous LRRK2 expression levels as another opportunity for independent corroboration of pRab10?

EMBOR: EMBOR-2024-60161V1

Dear Editors and Reviewers,

We appreciate the positive feedback and constructive advice provided for our manuscript "Microtubule regulation of LRRK2-mediated Rab phosphorylation." We are pleased that reviewers praised the quality, relevance and impact of the work.

"The work employs a rigorous application of modest exogenously expressed and endogenous LRRK2, and in some cases in the context of both WT and G2019S LRRK2. They find that microtubule depolymerization inhibits LRRK2 activation but the opposite is not true for microtubule stabilization. They importantly re-address reports of LRRK2 inhibitor induced association of LRRK2 protein with microtubules and data suggest these were largely artifacts of overexpression and tagging of LRRK2. While highly focused to interests of the LRRK2 community, the data are of high significance to that field given recent data provoking interest in LRRK2/microtubule interactions. There are only minor concerns for clarification."

Original Editor and Reviewer comments are pasted below followed by responses and the changes made to the manuscript.

Editor: 1. A data availability section providing access to data deposited in public databases is missing (where applicable).

Response: Raw data associated with this manuscript have been uploaded to the Zenodo repository [10.5281/zenodo.14549209](https://zenodo.org/doi/10.5281/zenodo.14549209)

Editor: 2. Your manuscript contains statistics and error bars based on $n=2$. Please use scatter plots in these cases.

Response: This has been fixed.

Editor: If the revision leads to a manuscript with more than 5 main figures it will be published as a Research Article. In this case the Results and Discussion section should be separate. If a Scientific Report is submitted, these sections have to be combined. This will help to shorten the manuscript text by eliminating some redundancy that is inevitable when discussing the same experiments twice. In either case, all materials and methods should be included in the main manuscript file.

Response: The article is a Research Article with 6 revised figures, separated Results and Discussion section

Editor 1) a .docx formatted version of the manuscript text (including legends for main figures, EV figures and tables). Please make sure that the changes are highlighted to be clearly visible.

Response: Changes are indicated in Dark Blue font, along with a clean version of the manuscript.

Editor: 2) individual production quality figure files as .eps, .tif, .jpg (one file per figure). See https://wol-prod-cdn.literatumonline.com/pb-assets/embo-site/EMBOPress_Figure_Guidelines_061115-1561436025777.pdf for more info on how to prepare your figures.

Response: All figures have been provided as high-resolution .tifs.

Editor: 3) We replaced Supplementary Information with Expanded View (EV) Figures and Tables that are collapsible/expandable online. A maximum of 5 EV Figures can be typeset. EV Figures should be cited as 'Figure EV1, Figure EV2" etc... in the text and their respective legends should be included in the main text after the legends of regular figures. Additional Tables/Datasets should be labeled and referred to as Table EV1, Dataset EV1, etc. Legends have to be provided in a separate tab in case of .xls files. Alternatively, the legend can be supplied as a separate text file (README) and zipped together with the Table/Dataset file.

Response: Cited EV figures are as Fig. EV1, Fig. EV2-EV5

Editor: - For the figures that you do NOT wish to display as Expanded View figures, they should be bundled together with their legends in a single PDF file called *Appendix*, which should start with a short Table of Content. Appendix figures should be referred to in the main text as: "Appendix Figure S1, Appendix Figure S2" etc. See detailed instructions regarding expanded view here: <https://www.embopress.org/page/journal/14693178/authorguide#expandedview>;

Response: We include an Appendix Figure S1-3.

Editor: 4) a .docx formatted letter INCLUDING the reviewers' reports and your detailed point-by-point responses to their comments. As part of the EMBO publication's Transparent Editorial Process, EMBO reports publishes online a Review Process File (RPF) to accompany accepted manuscripts. This File will be published in conjunction with your paper and will include the referee reports, your point-by-point response and all pertinent correspondence relating to the manuscript.

<https://www.embopress.org/page/journal/14693178/authorguide#transparentprocess>

Response: Included.

Editor: 5) a complete author checklist, which you can download from our author guidelines <https://www.embopress.org/page/journal/14693178/authorguide>. Please insert information in the checklist that is also reflected in the manuscript. The completed author checklist will also be part of the RPF.

Response: Included.

Editor: 6) Please note that all corresponding authors are required to supply an ORCID ID for their name upon submission of a revised manuscript (<<https://orcid.org/>>). Please find instructions on how to link your ORCID ID to your account in our manuscript tracking system in our Author guidelines <<https://www.embopress.org/page/journal/14693178/authorguide#authorshipguidelines>>;

Response: Done.

Editor: 7) Before submitting your revision, primary datasets produced in this study need to be deposited in an appropriate public database (see <https://www.embopress.org/page/journal/14693178/authorguide#datadeposition>). Please remember to provide a reviewer password if the datasets are not yet public. The accession numbers and database should be listed in a formal "Data Availability" section placed after Materials & Method (see also <https://www.embopress.org/page/journal/14693178/authorguide#datadeposition>). Please note that the Data Availability Section is restricted to new primary data that are part of this study. * Note - All links should resolve to a page where the data can be accessed. * If your study has not produced novel datasets, please mention this fact in the Data Availability Section.

Responses: Deposited at Zenodo DOI:10.5281/zenodo.14549209 This fact has also been mentioned in the "Data Availability Section" in the manuscript.

Editor: 8) At EMBO Press we ask authors to provide source data for the main figures. Our source data coordinator will contact you to discuss which figure panels we would need source data for and will also provide you with helpful tips on how to upload and organize the files.

Additional information on source data and instruction on how to label the files are available: <https://www.embopress.org/page/journal/14693178/authorguide#sourcedata>

Response: Source data are deposited at Zenodo DOI:10.5281/zenodo.14549209

Editor: Editor: 10) Regarding data quantification (see Figure Legends: <https://www.embopress.org/page/journal/14693178/authorguide#figureformat>)

The following points must be specified in each **figure legend**:

The name of the statistical test used to generate error bars and P values,- the number (n) of independent experiments (please specify technical or biological replicates) underlying each data point, the nature of the bars and error bars (s.d., s.e.m.), Each legend contains the statistical test used.- Please also include scale bars in all microscopy images.

Response: Each legend contains all the requested information.

Editor: 11) The journal requires a statement specifying whether or not authors have competing interests (defined as all potential or actual interests that could be perceived to influence the presentation or interpretation of an article). In case of competing interests, this must be specified in your disclosure statement. Further information:

<https://www.embopress.org/competing-interests>

Response: The statement is included.

Editor: 12) Please also note our reference format:

Response: References have been appropriately formatted

Editor: 13) All Materials and Methods need to be described in the main text using our 'Structured Methods' format, which is required for all research articles. According to this format, the Methods section includes a Reagents and Tools Table (listing key reagents, experimental models, software and relevant equipment and including their sources and relevant identifiers) followed by a Methods and Protocols section describing the methods using a step-by-step protocol format. The aim is to facilitate adoption of the methodologies across labs. More information on how to adhere to this format as well as a downloadable template (.docx) for the Reagents and Tools Table can be found in our author guidelines:

Response: Structure Method format is included for the Materials and Methods Section.

Referee #3: This manuscript explores the consequences of microtubule stabilization and fragmentation on LRRK2 localization and activity. The work employs a rigorous application of modest exogenously expressed and endogenous LRRK2, and in some cases in the context of both WT and G2019S LRRK2. They find that microtubule depolymerization inhibits LRRK2 activation but the opposite is not true for microtubule stabilization. They importantly re-address reports of LRRK2 inhibitor induced association of LRRK2 protein with microtubules and data

suggest these were largely artifacts of overexpression and tagging of LRRK2. While highly focused to interests of the LRRK2 community, the data are of high significance to that field given recent data provoking interest in LRRK2/microtubule interactions. There are only minor concerns for clarification.

Response: We thank the reviewer for these comments and for the time and effort in helping us to further improve impact.

Referee #3: 1. For Fig 1 There is really no unaffected loading control protein for the western blots as LRRK2, Rab10, and tubulin are all affected in some way by the various treatments. Actin may be unwise later on as cytoskeleton dynamics are purposefully manipulated.

Response: Revised Fig. 1 now prioritizes measures of the ratio of pRab10 to total Rab10 as an objective measure that does not need normalization to cytoskeleton proteins. Of note, we have not observed any of the manipulations to consistently influence total Rab10 protein, consistent with our past work with Rab10 protein in many different cells and tissue types (PMID 38862989, 38862989, and 35012605).

Referee #3: 2. G2019S is a weak Rab kinase mutation - have the investigators looked at R1441C or Y1699C? These also differ in 14-3-3 binding.

Response: We now include Figure EV2 that shows nocodazole attenuates the ratio of pRab10 to total Rab10 in transfections of LRRK2 and mutated (R1441C, Y1699C) LRRK2. While nocodazole's effects in the HEK293 and mutant LRRK2 system are not quite as dramatic on pRab10 levels as in macrophages and A549 cells, the trends are preserved. Unfortunately we did not have access to macrophages with these pathogenic LRRK2 mutations during the revision period, but the newly included data extends to core observations of this study to another orthogonal cell model in the HEK293T cells that support the main conclusions.

Referee #3: 3. The Docetaxel images in 1E do not reflect those in 1A. Clearer effects are seen in 1E.

Response: We agree and now created a new Fig. EV4 that includes larger docetaxel images easier to visualize, since more space is now allotted in the new EV figureset allowed by the Editors than previously allowed with the main figures. We can also include additional blow-ups of single microtubules in new Appendix files, as all scans were from AiryScan high-resolution technology, if that would be helpful to reviewers and readers, but we think the new images included should suffice.

Referee #3: 4. There is a focus on Rab10 and pRab10 in the context of basal LRRK2 activity, lysosomal injury and membrane recruitment. Does one see the same effects with pRab8a (despite antibody specificity limitations) as it is likewise implicated in inflammatory responses and lysosomal injury?

Response: We agree with the reviewer on our current focus, but in consultation with the antibody team at MJFF, the only pRab8a antibody recommended are clones that cross-react with pRab10, so the results with these clones might be difficult to interpret. Though there might be ways around this issue, for example mass spectrometry or co-immunoprecipitations in capture and detect strategies, these alternative approaches have been very difficult for us to optimize in the reliable detection of endogenous pRab8a (which is much lower than pRab10 in our cell models). However, the revised version presents new pRab10 results in several new human cell models (A549 and HEK293Ts) as well as more data from mouse macrophage cell lines.

Referee #3: Can pS1292 be applied here for high endogenous LRRK2 expression levels as another opportunity for independent corroboration of pRab10?

Response: We agree that pS1292-LRRK2 data would have provided potentially valuable insights regarding LRRK2 activity and activation. Throughout our studies, from BMDMs in Fig. 1 to A549 and RAW macrophage cell lines, we attempted to blot the lysates utilized for pRab staining with pS1292-LRRK2 antibodies from Abcam (MJFR19). However, the current lots of antibodies we received from Abcam were of insufficient quality to detect signals easily discernible from off-target bands very close to the expected size of LRRK2, in all of these cell systems. We contacted MJFF tools program about this issue, since past lots of MJFR19 were much more sensitive and specific, and were told that Abcam has been unable to improve the sensitivity and specificity of the clones from what is currently available. An example of even transfected HEK-293T is shown below, where endogenous pRab10 is strongly activated but the pS1292-LRRK2 antibody only detects signal independent of LRRK2 expression (even when transfected). Our lab has published many papers measuring pS1292-LRRK2, both endogenous and from transfected protein, so this is an unfortunate setback to the LRRK2 field.

Though the current pS1292-LRRK2 antibodies from Abcam are effective at measuring pS1292-LRRK2 in recombinant LRRK2 protein preparations, we would suggest caution to those groups who wish to incorporate this antibody into different physiologically-relevant experimental systems.

Referee #1:

Referee #1: This generally, well written manuscript reports the surprising observation that LRRK2 phosphorylation of Rab10, Rab8 and Rab35 is strongly decreased in nocodazole-treated bone marrow derived macrophages, a condition in which they contain depolymerized microtubules. Nocodazole also interferes with pRab10 generation upon lysosome stress in these cells. Using streptolysin to add GTP to cells, the authors see less phosphorylation but it is not clear if this is important for LRRK2, Rab substrates or both. Locking LRRK2 on lysosomes enhances Rab phosphorylation, likely because the Rab is mistargeted there and cannot be retrieved by GDI; similar results have been reported by others upon organelle mistargeting in other cell types (Gomez). The authors conclude that nocodazole interferes with this lysosome activation but they may be using an invalid metric to conclude this.

Response: We thank the reviewer for this careful analysis and the time and effort placed reviewing our work. We eliminated the metrics in question as described below, and the recommended approaches produce the same conclusions as previously described.

Referee #1: They confirm known protein proximities using APEX and conclude that bulky tags on LRRK2 increase the chances of inhibitor-driven aggregation that is non-physiological.

Referee #1: The story would be of great interest to readers of EMBO Reports if the authors probed the mechanism(s) of their unexpected nocodazole findings. There are several possible explanations. One is that nocodazole activates the Rab PPM1H phosphatase (known to prefer highly curved membranes that would be generated by Golgi fragmentation upon nocodazole hasten addition). To test if nocodazole activates the Rab phosphatase, the authors should monitor the rate of dephosphorylation of phosphoRab10 from time zero of nocodazole addition by adding MLI-2 at time zero and tracking rate of phosphoRab10 decrease for various times.

Response: We thank the reviewer for the suggestion of this experiment. In primary macrophages, we find that pRab10 levels drop quite quickly, within minutes, after MLI2 or nocodazole addition. The data now included in the revision clearly shows that nocodazole does not hasten the loss of pRab10 after MLI2-mediated LRRK2 inhibition, suggesting nocodazole does not activate PPM1H (Figure EV3). We were surprised that nocodazole works as quickly as it does on reducing pRab10 to total Rab10 ratios, for example within 5 minutes. Though very unlikely, we tested whether nocodazole is a LRRK2 kinase inhibitor using our standard *in vitro* LRRK2 kinase assays. There was no reduction of LRRK2 kinase activity at all *in vitro* up to 10 micromolar nocodazole, as expected, so the MOA of nocodazole is indirect.

Reviewer #1: Another possibility is that nocodazole moves Rab12-activating vesicles away from those harboring Rabs 8, 10 or 35, leading to less LRRK2 activation. In this latter scenario, MLI-2 sensitive Rab12 phosphorylation would not change and this can easily (and should) be tested.

Response: We thank the reviewer for this suggestion. Unfortunately, the pRab12 antibodies we have access to and recommended by MJFF for the detection of pRab12 did not yield any

specific signal either by immunoblot or immunocytochemistry in our cell models. A representative blot is pasted below. That nocodazole has effects within minutes on the ratio of pRab10 to total Rab10, and can be rescued by GTP supplementation, raises the possibility that nocodazole-induced GTP-depletion gradients may be adversely affecting LRRK2 and Rab12 activation in cells. However, consistent with a long-standing limitation in the microtubule field, there is no direct way to visualize rapid GTP or GDP gradients in living cells at the necessary

spatial and temporal resolutions that would be required.

Reviewer #1: A major limitation of some of the analysis herein is a metric that the authors use of a "Rab10 vesicle" [Fig. 1G,H, 2 B,D,I, 4C,D: "pRab10 vesicles per cell"] At the EM level (and higher resolution light microscope level) there are surely more than 1.5 pRab10 vesicles-these cells contain ~2.5 million copies of Rab10, about half of which are membrane bound. In any case, it would be more correct to determine the total intensity of anti-pRab10 staining since the number of pRab10 structures don't serve as a useful indicator of total kinase activity and the value is not meaningful in relation to actual vesicle numbers. Again, 50 vesicles of a size below the diffraction limit, concentrated at the centrosome would look like one or two larger structures-and are harder to detect once dispersed by nocodazole. These graphs must all be corrected.

Response: We thank the review for the suggestions and agree with the comments. We were following a protocol we previously developed to measure a subclass of larger Rab10 vesicles demarcated by dextran (PMID 32853409). In this study, we do not include dextran, as the

interest is more on measuring pRab10 (and total Rab10 intensity) in cells. Therefore, all graphs have been corrected in this manner. Of note, the conclusions and group differences remain virtually the same with the new analyses, likely because the concentrations of pRab10 and total Rab10 on the larger vesicles are strongly correlated to total signal intensities.

Additional comments:

Reviewer #1: The intro is written in a very scholarly manner

Reviewer #1 Figure 1. The blots in B and C are analyzed using non-linear ECL which will exaggerate and saturate signals. For example, Rab10 is highly overexposed and there is much less in lanes 5-8 in the wild type example. Lighter exposures of tubulin and Rab10 need to be shown or better yet, use of a fluorescent second antibody. In any case, nocodazole has a clear impact. (panels G and H are invalid).

Response: We use a digital Chemidoc system from Biorad that records 16-bit .scn files that proactively flag any image with evidence of saturation (e.g., overexposed), and these are not utilized for quantification. The darker exposures were attempts to simultaneously display remnant pRab10 signal after nocodazole exposures instead of blank lanes, and then other blots and targets were selected also with dark bands to match the target pRab10 blots. However, the point is well taken, and we routinely 'challenge' the linearity of the approach during different studies. We now include one such quality control run as Figure EV1A,B, which demonstrates linearity of intensity across pixel intensities used in this study. Though the bands appear fully black in print, in 16-bit mode analysis of the .scn files, none of the information is over-exposed (i.e., fully black). Further, as detailed in the revised methods, quantifications are performed automatically by software operated by investigators blinded to the lane identity.

Reviewer #1 The authors state, "...there was an obvious reduction in Rab vesicle localization after nocodazole"...."stabilized in a GDI-bound cytoplasmic form" This has not been shown. To make this statement, the authors need to carry out membrane-cytosol cell fractionation. Dispersal of small (100K G X 60 minute sedimentable) vesicles is quite different than a change in Rab protein membrane association.

Response: We agree, and this statement has been removed. During the revision period, we attempted several techniques to separate membranes from cytosols in macrophages, and in every case, we lost nearly all pRab10 signal, without acceptable fractionation. With the new data in Figure EV3 showing MLI2 effects within minutes of addition to the cells, we suppose that macrophages have very high levels of PPMH1 in addition to other proteases that are difficult to fully and reliably inhibit in extensive post-lysis processing steps.

Reviewer #1 Figure 2. B, D, I are invalid; not sure what we learn from GTP addition experiment.

Response: Previous figures 2B/D that focused on quantification of vesicles have been removed. The GTP addition experiment was recommended to us by experts working on microtubules in the neurodegeneration field (personal communications with George S. Bloom), and we were surprised with the high efficiency of recovery of nocodazole phenotypes with strong GTP supplementation. We verified recovery both by immunocytochemistry and

immunoblots, the main workhorses of our quantitative assessments, in revised Fig. 2G,H. The feedback we have so far received from other experts in the microtubule field are positive with this experiment, with the notion that many phenotypes that rapidly induce with nocodazole treatment might be due to GTP/GDP gradient effects, but directly measuring this is technically challenging as mentioned. We cannot rule out other possibilities. However, the point from the reviewer is taken and we do not highlight the results from the GTP addition experiment in the abstract or main conclusions, as convincingly resolving the mechanisms of action of nocodazole, while of interest, were not included in the main rationale for this particular study.

Reviewer #1 Figure 3. Please label the tags in relation to their N- or C-terminal locations in the constructs: It should be LAMP1-FRB for example.

Response: Done

Reviewer #1: Does nocodazole have any effect in cells other than macrophages? Others seem not to see the nocodazole effect in LRRK2 R1441 MEFs (see Figure 6A, DOI: 10.1126/sciadv.adj1205). If macrophage-specific, please state this.

Response: As mentioned, revised Fig. 1F includes the analysis of A549 cells that we and others have shown with reasonable endogenous LRRK2 and Rab10 levels, and found relationships similar to mouse macrophages, human macrophages, mouse RAW-264 cells, and transfected HEK-293T cells. In most experiments from these different kinds of cells, nocodazole treatment does not fully ablate pRab10 levels like MLi2, potentially allowing for Fig. 6A in DOI: 10.1126/sciadv.adj1205.

Reviewer #1 Please indicate in legend where reader can find blots for panel E.

Response: Blots are now included in Fig. 3E,G

Reviewer #1 Figure 4. Constructs in this figure should be labeled "FKBP-". Images need to re-quantified and #vesicles per cell removed (panels C and D).

Response: Constructs in revised Fig 3 and 4 are now labeled appropriately with FKBP-, and images have been re-quantified (revised Fig 4C,D) with #vesicles per cell removed.

Reviewer #1 Figure 5. As the authors note, 90% of LRRK2 is cytosolic. That tubulin is labeled by APEX is not a surprise since tubulin represents a huge fraction of total cellular proteins. Proximity to gamma tubulin confirms centriolar proximity, and APEX labeling of Rab29 (7L1), 10 and 35 are expected. Not sure what more one can conclude from this figure, despite the clean data. Since Rab8A, 10 and 35 bind to the LRRK2 ARM domain, insensitivity to MLi-2 is also not unexpected. Microscopy from many labs shows aggregates of overexpressed, inhibited LRRK2 along microtubules-but the reviewer agrees that this is non-physiological.

Response: We agree with all of these points, and highlight the purpose of Fig. 5 which uses the APEX2 labeling method to better understand LRRK2 association with tubulin after treatment

with nocodazole and docetaxel. Prior literature suggests that LRRK2 has preferential affinity towards microtubules. Therefore, it might be presumed that docetaxel would increase the amount of biotinylated tubulin from APEX2-labeled LRRK2, whereas nocodazole would decrease LRRK2-labeled tubulin (especially when bound to MLI2). The main conclusions do not support these presumptions, since LRRK2 preferentially maintains proximity to nearby tubulins independent of tubulin polymerization. These results, together with other datasets mentioned by the Reviewer, collectively argue against stable and high-affinity interactions between LRRK2 and microtubules.

Reviewer #1 Figure 6. Here the authors conclude that large tags increase apparent microtubule-associated MLI-2 induced aggregation not seen in brain. Not sure how much one can conclude from changes in serum levels of LRRK2 {plus minus}MLI-2 (panel D)? The brain images show one neuron with LRRK2 signal. Detection of LRRK2 in brain is important (here also nicely controlled) and authors should quantify staining over 100 cells of a particular type to provide information related to LRRK2 distribution and distribution of cellular intensities. Really not sure why this data is included in this paper.

Response: We thank the reviewer for pointing out that these experiments needed additional clarification and explanation, as well as additional quantification. These experiments were in response to earlier data (circa 2010) demonstrating a proportion of LRRK2 protein associated with microtubules in neurons based on immunoEM (PMCID: PMC2906661), and more recent data suggesting the interactions between LRRK2 and microtubules (at least in HEK293T cells) would be greatly increased with MLI2 exposure (e.g., PMCID: PMC7726071). We wondered whether LRRK2 would increase in MT association in response to the brain-penetrant MLI2 molecule. We scanned LRRK2 immunohistochemical distribution throughout the entire mouse brain and did not observe any changes between the relative association of LRRK2 protein and microtubule alignments (e.g., LRRK2-positive skeins). We now present, in revised Fig. 6, quantification from LRRK2 colocalization with neuronal-specific Tuj1 from both the motor cortex and striatum. We feel these data collectively add to the conclusion that LRRK2 positive skeins are not physiologically relevant structures, even in mature cells in the brain with extensive microtubule networks that express the highest levels of LRRK2.

We thank both Reviewers and Editors for their help in improving our manuscript.

Sincerely,

Andy West and

Tuyana Malankhanova

Dear Andy,

Thank you for submitting your revised manuscript. It has now been seen by both of the original referees. I apologize for this unusual delay in getting back to you. As previously communicated, it took longer than anticipated to receive the referee reports.

As you can see, both referees find that the study is significantly improved during revision and recommend publication. Please note that referee #3 did not provide a report, but contacted us to let us know of his/her support for publication of the revised manuscript. However, I need you to address the points below before I can accept the manuscript.

- Please address the remaining concerns of referee #1 (textually) and provide a response.
- Please remove the 'Author Contributions' section from the manuscript text.
- Please submit the author checklist in excel format.
- Please make sure that funding information is complete in both the manuscript submission system and the manuscript text. Currently, we note that funding information not mentioned in the manuscript; there is one funder entered in the manuscript submission system, but it is missing in the manuscript.
- We note that the legends and in-text callouts refer to a Figure EV5, but there are currently 4 EV figures provided.
- We note that the following figure panels are currently not called out in the text: Fig. 5E, Fig. 5F, Fig. 6D.
- Please add page numbers to the Appendix file.
- Please remove the Reagents and Tools Table from the manuscript text and submit it as a separate file.
- In the Data Availability section, please provide a link that directly resolves to the source data hosted at zenodo.org. Also, please fill in the Source Data checklist (attached).
- During our routine source data checks, in the Fig 2B excel file, we note that all three replicates of the mli2 condition have the same numerical value (attached). Please clarify.
- Again, during our routine checks, we note some potential image reuses between Figure 3 A,B,C & D and Appendix Figure S1B. Please clarify, also in all respective figure legends.
- Our production/data editors have asked you to clarify several points in the figure legends:
 - o Please note that the legend for figure EV3 C is missing in the manuscript. This needs to be rectified.
 - o Please note that the exact p values are not provided in the legends of figures 1C, E, F, G; 2B, C, E, F, G, H; 3F, H; 4C, D; 5B, 6B, EV2 B.
 - o Please indicate what * / ** / *** / **** represents; if this represents p value(s), please indicate the statistical test used and the exact p value in the legend(s) of figure(s) EV4 B.
 - o Please note that in figure EV2 B there is a mismatch between the annotated p values in the figure legend and the annotated p values in the figure file that should be corrected.
 - o Please note that information related to n is missing in the legends of figures 3H, EV4 B.
 - o Please note that the error bars are not defined in the legend of figure EV4 B.
 - o Please note that the scale bar needs to be defined for figures EV3A-C.
- Papers published in EMBO Reports include a 'synopsis' and 'bullet points' to further enhance discoverability. Both are displayed on the html version of the paper and are freely accessible to all readers. The synopsis includes a short standfirst summarizing the study in 1 or 2 sentences (max 35 words) that summarize the paper and are provided by the authors and streamlined by the handling editor. I would therefore ask you to include your synopsis blurb and 3-5 bullet points listing the key experimental findings.
- In addition, please provide an image for the synopsis. This image should provide a rapid overview of the question addressed in the study but still needs to be kept fairly modest since the image size cannot exceed 550 (width) x 300-600 (height) pixels.

Thank you again for giving us to consider your manuscript for EMBO Reports, I look forward to your minor revision.

Kind regards,

Deniz

--

Deniz Senyilmaz Tiebe, PhD
Senior Scientific Editor
EMBO Reports

Referee #1:

First, thanks to the authors for their thoughtful responses to the reviewer comments. Their edits give me an entirely new and different picture of what is being studied herein and how to think about it. The paper reports the unexpected and striking finding

that nocodazole depolymerization of microtubules in cells decreases LRRK2 activity, especially in macrophages but also a bit in A549 cells and maybe 293 cells. If this is because microtubules stimulate LRRK2, it should go up with taxol but it does not. Indeed, the title is "Microtubule regulation of LRRK2-mediated Rab phosphorylation" but no microtubule regulation is shown here-that would involve specific residues on LRRK2 binding to microtubules and leading to kinase activation-instead they see a constant level of microtubule association, independent of kinase activity. (The title must be changed--see below)

A really important key experiment is their use of streptolysin O and addition of GTP which completely restores LRRK2 activity. Cool! This says that GTP is the regulator, not microtubules, since this is the only thing that changes in that experiment. The simplest explanation for what is going on is that tubulin depolymerization is sequestering GDP on tubulin subunits, decreasing cytosolic GTP levels needed for LRRK2 activity. Nocodazole leads to the accumulation of tubulin in the GDP-bound state, which cannot directly exchange bound GDP for use in GTP generation. This decreases the cellular availability of free GTP. Nocodazole may thus decrease the cytoplasmic GTP pool, thereby blocking LRRK2 activity.

Tubulin is 10-20 μ M. Studies suggest that 1-5% of the total GTP pool is free in the cytoplasm. For a typical cellular GTP concentration of 200-500 μ M, this translates to 2-25 μ M of free GTP; close to that of total tubulin.

Thus, a better title and conclusion would be: "Exquisite sensitivity of LRRK2 kinase action to cellular GTP". This is a surprising and important result. The authors could supplement intact cells with guanosine or inosine to support this conclusion; also there might be a difference in sensitivity for ROC-COR mutations versus kinase domain mutations. True, the authors would not yet have proven that it was LRRK2 and not the Rabs that were sensitive, but Rab10 has a very high affinity for nucleotide so it is much more likely the LRRK2 that doesn't have enough GTP for activation. [Note that other labs may also get different results depending on their growth media content of guanosine or glucose and cell confluency-the authors should note this.]

Finally, in the abstract and discussion the authors state that nocodazole impairs Rab membrane localization but that is not shown here-they see only dispersed Rab10 staining (Fig. 1D,E) which will always be harder to visualize in non-adherent cells, equally consistent with its presence in small vesicles. To make that conclusion they would need to look at membrane versus cytosol pools which for some reason did not work in their hands; that conclusion must be removed until shown directly.

I support presentation of the work with a new title and rephrasing of the findings in line with the most likely explanation and caveats discussed here.

Minor comment: is "drug exposure" in EV3 is for just Mli-2 or for both Mli-2 and nocodazole?

EMBOR: EMBOR-2024-60161V2

Dear Editors and Reviewers,

We appreciate the positive minor revisions requested to improve our manuscript now titled “LRRK2 interactions with microtubules are independent of LRRK2-mediated Rab phosphorylation.” Original Editor and Reviewer comments are pasted below followed by responses and the corresponding changes to the manuscript.

Editor 1: Please remove the 'Author Contributions' section from the manuscript text.

Response: This has been done

Editor 2: Please submit the author checklist in excel format.

Response: This has been submitted.

Editor 3: Please make sure that funding information is complete in both the manuscript submission system and the manuscript text. Currently, we note that funding information not mentioned in the manuscript; there is one funder entered in the manuscript submission system, but it is missing in the manuscript.

Response: Funding information has been added to the submission system and manuscript text.

Editor 4: We note that the legends and in-text callouts refer to a Figure EV5, but there are currently 4 EV figures provided.

Response: Figure EV5 has been provided.

Editor 5: We note that the following figure panels are currently not called out in the text: Fig. 5E, Fig. 5F, Fig. 6D.

Response: These panels are now called out in the text.

Editor 6: Please add page numbers to the Appendix file.

Response: This has been done.

Editor 7: Please remove the Reagents and Tools Table from the manuscript text and submit it as a separate file.

Response: This has been done. The table has been added as “Appendix Table S1” as part of the Appendix file. This file is now called out in the revised Methods section

Editor 8: In the Data Availability section, please provide a link that directly resolves to the source data hosted at zenodo.org.

Response: The Zenodo link is now provided in the Data Availability section corresponding to the DOI 10.5281/zenodo.15206770 (<https://zenodo.org/records/15206770>)

All source data .zip files corresponding to each figure are provided through this resource.

Editor 8 continued: Also, please fill in the Source Data checklist (attached).

Response: This form is filled and now uploaded. As mentioned above, all source data are accessible via the Zenodo link (<https://zenodo.org/records/15206770>)

Editor 9: During our routine source data checks, in the Fig 2B excel file, we note that all three replicates of the mli2 condition have the same numerical value (attached). Please clarify.

Response: This error has been fixed.

Editor 10: Again, during our routine checks, we note some potential image reuses between Figure 3 A,B,C & D and Appendix Figure S1B. Please clarify, also in all respective figure legends.

Response: The reuse is intentional as a high-magnification expansion of lower magnification images, highlighting critical features as requested by Reviewers. This point has been clarified in the legends with respect to images in Appendix Figure S1B which are high magnification images of Figure 3 A,B,C & D.

Editor 11: Please note that the legend for figure EV3 C is missing in the manuscript. This needs to be rectified.

Response: This has been added.

Editor 12: Please note that the exact p values are not provided in the legends of figures 1C, E, F, G; 2B, C, E, F, G, H; 3F, H; 4C, D; 5B, 6B, EV2 B.

Response: Exact p values are now provided in these legends.

Editor 13: Please indicate what */ **/ ***/ **** represents; if this represents p value(s), please indicate the statistical test used and the exact p value in the legend(s) of figure(s) EV4 B.

Response: This has been updated as requested.

Editor 14: Please note that in figure EV2 B there is a mismatch between the annotated p values in the figure legend and the annotated p values in the figure file that should be corrected.

Response: This has been corrected.

Editor 15: Please note that information related to n is missing in the legends of figures 3H, EV4 B.

Response: This has been updated.

Editor 16: Please note that the error bars are not defined in the legend of figure EV4 B.

Response: This has been updated.

Editor 17: Please note that the scale bar needs to be defined for figures EV3A-C.

Response: This has been updated as “± SEM error bars”.

Editor 18: Papers published in EMBO Reports include a 'synopsis' and 'bullet points' to further enhance discoverability. Both are displayed on the html version of the paper and are freely accessible to all readers. The synopsis includes a short standfirst summarizing the study in 1 or 2 sentences (max 35 words) that summarize the paper and are provided by the authors and streamlined by the handling editor. I would therefore ask you to include your synopsis blurb and 3-5 bullet points listing the key experimental findings

Response: The synopsis and x4 bullet points have been added to the manuscript text.

Editor 19: In addition, please provide an image for the synopsis. This image should provide a rapid overview of the question addressed in the study but still needs to be kept fairly modest since the image size cannot exceed 550 (width) x 300-600 (height) pixels.

Response: An image has been uploaded at 550x600 pixel width as “Synopsis Image”.

Referee #1: First, thanks to the authors for their thoughtful responses to the reviewer comments. Their edits give me an entirely new and different picture of what is being studied herein and how to think about it. The paper reports the unexpected and striking finding that nocodazole depolymerization of microtubules in cells decreases LRRK2 activity, especially in macrophages but also a bit in A549 cells and maybe 293 cells. If this is because microtubules stimulate LRRK2, it should go up with taxol but it does not. Indeed, the title is "Microtubule regulation of LRRK2-mediated Rab phosphorylation" but no microtubule regulation is shown here-that would involve specific residues on LRRK2 binding to microtubules and leading to kinase activation-instead they see a constant level of microtubule association, independent of kinase activity. (The title must be changed--see below).

Response: We agree, and updated the title to "LRRK2 interactions with microtubules are independent of LRRK2-mediated Rab phosphorylation". This title better captures the hypotheses we set out to test at study initiation, and we believe fairly summarize the key points reflected in the abstract.

Referee #1: A really important key experiment is their use of streptolysin O and addition of GTP which completely restores LRRK2 activity. Cool! This says that GTP is the regulator, not microtubules, since this is the only thing that changes in that experiment.

Response: We thank the reviewer for pointing this out, and we have updated the Discussion section to include this observation.

Referee #1: The simplest explanation for what is going on is that tubulin depolymerization is sequestering GDP on tubulin subunits, decreasing cytosolic GTP levels needed for LRRK2 activity. Nocodazole leads to the accumulation of tubulin in the GDP-bound state, which cannot directly exchange bound GDP for use in GTP generation. This decreases the cellular availability of free GTP. Nocodazole may thus decrease the cytoplasmic GTP pool, thereby blocking LRRK2 activity. Tubulin is 10-20 μ M. Studies suggest that 1-5% of the total GTP pool is free in the cytoplasm. For a typical cellular GTP concentration of 200-500 μ M, this translates to 2-25 μ M of free GTP; close to that of total tubulin.

Response: We thank the reviewer for these observations that help us interpret the data and will assuredly help inform readers. We have updated the Discussion section to include this results with pertinent additional references.

The excerpt below is now present in the revised discussion:

"This study finds that microtubule destabilization via nocodazole treatment may impair Rab activation and LRRK2-mediated Rab phosphorylation in macrophages and other cell types, effects that can be recovered with the supplementation GTP. Tubulin concentration varies between 10-20 μ M in mammalian cells (Shida et al, 2010; Hiller & Weber, 1978). The average total GTP concentration in cells is 500 μ M (Traut, 1994), whereas the total free GTP pool in cytoplasm is around 30 μ M or less (Wolff et al, 2022), close to estimated concentrations of free tubulin in microtubule-absent cells. Tubulin depolymerization by nocodazole leads to the rapid accumulation of free tubulin in the GDP-bound state thereby limiting the production of GTP. Consequently, nocodazole may transiently reduce the cytoplasmic GTP pool, which may affect local LRRK2 GTP-binding potentially required for LRRK2-mediated Rab phosphorylation. Ours and others past studies show that LRRK2 affinity for GTP is low (Liu et al, 2016; Liu & West, 2016) compared to Rab proteins (Dumas et al, 1999; Simon et al, 1996), suggesting GTP-

supplementation experiments may rescue nocodazole effects through upregulating GTP-bound LRRK2 protein. Though methods to measure GTP-bound LRRK2 in living cells have not yet been described, future studies that directly measure cellular free GTP levels as they may direct Rab substrate phosphorylation may be of interest in exploring rapid stress-activation of LRRK2 phosphorylation pathways.”

Referee #1: Thus, a better title and conclusion would be: "Exquisite sensitivity of LRRK2 kinase action to cellular GTP".

Response: We revised the title as described above, and added a new sentence (pasted below) to the abstract that might better characterize the current conclusions of the study:

“GTP supplementation restores nocodazole-reduced Rab phosphorylation, linking LRRK2 kinase action to cellular GTP levels.”

Referee #1: This is a surprising and important result. The authors could supplement intact cells with guanosine or inosine to support this conclusion; also there might be a difference in sensitivity for ROC-COR mutations versus kinase domain mutations. True, the authors would not yet have proven that it was LRRK2 and not the Rabs that were sensitive, but Rab10 has a very high affinity for nucleotide so it is much more likely the LRRK2 that doesn't have enough GTP for activation. [Note that other labs may also get different results depending on their growth media content of guanosine or glucose and cell confluency-the authors should note this.]

Response: We sincerely thank the reviewer for elaborating this important line of investigation. First, we have added these thoughts in the revised Discussion section regarding the need for targeted additional experimentation in the manipulation of cellular GTP levels with respect to ROC-COR mutations vs. kinase domain mutation in Rab phosphorylation. Indeed, we can envisage that cellular GTP levels may be key to ‘unmasking’ the effects of certain pathogenic mutations on Rab substrate phosphorylation depending on the model. The following paragraph has been added to the discussion:

“This study finds that microtubule destabilization via nocodazole treatment may impair Rab activation and LRRK2-mediated Rab phosphorylation in macrophages and other cell types, effects that can be recovered with supplemented GTP. Tubulin concentration varies between 10-20 μM in mammalian cells (Shida et al, 2010; Hiller & Weber, 1978). The average total GTP concentration in cells is 500 μM (Traut, 1994), whereas the total free GTP pool in cytoplasm is around 30 μM or less (Wolff et al, 2022), therefore close to estimated concentrations of free tubulin in microtubule-absent cells. Tubulin depolymerization by nocodazole leads to the rapid accumulation of free tubulin in the GDP-bound state thereby limiting the production of GTP. Consequently, nocodazole may transiently reduce the cytoplasmic GTP pool, which may affect LRRK2 GTP-binding potentially required for LRRK2-mediated Rab phosphorylation. Ours and others past studies show that LRRK2 affinity for GTP is low (Liu et al, 2016; Liu & West, 2016) compared to Rab proteins (Dumas et al, 1999; Simon et al, 1996), suggesting GTP-supplementation experiments may rescue nocodazole effects through upregulating GTP-bound LRRK2 protein. Though methods to measure GTP-bound LRRK2 in living cells have not yet been described, future studies that directly measure cellular free GTP levels as they may direct Rab substrate phosphorylation may be of interest in exploring rapid stress-activation and maintenance of LRRK2 phosphorylation pathways.”

During the revision period, we performed some initial experiments with guanosine or inosine supplementation in mouse primary macrophages. We were excited that either guanosine (100µM, 30 min) or inosine (100µM, 30 min) rapidly increases pRab10 levels in the cells. However, the effect on recovery of nocodazole effects were not clear over several experiments (see panel A-C to the right), and results were similar in A549 cells (not shown here). The main issue is we lack, at the moment, a reliable technique to measure free GTP levels accurately in soluble lysates from these cells. To our surprise, we were unable to locate a working HPLC suitable for rapid GTP measurements. The GC/MS instrumentation available in our metabolomics core can measure GTP, but we are not confident in quantification yet in the lysates. It seems GTP is surprisingly unstable and rapidly degrades to GDP and GMP in our lysates, which presents difficulties in working with off-site mass spectrometry instrumentation. Because we are not yet confident on which conditions of inosine and guanosine exposure leads to increases of cellular GTP, especially during nocodazole treatment, we think the results we have so far are inconclusive, but compelling enough to continue on in this area for prioritized future study. Further, we are nearing both the limits of the number of figures and the text for this current report, as well as revision period duration.

We sincerely thank the reviewer for this suggestion and helping to guide our experimentation.

Referee #1: Finally, in the abstract and discussion the authors state that nocodazole impairs Rab membrane localization but that is not shown here-they see only dispersed Rab10 staining (Fig. 1D,E) which will always be harder to visualize in non-adherent cells, equally consistent with its presence in small vesicles. To make that conclusion they would need to look at membrane versus cytosol pools which for some reason did not work in their hands; that conclusion must be removed until shown directly.

Response: We agree, and removed all statements related to “nocodazole impairing Rab membrane localization”, from the abstract and text. We think these revisions do not significantly impact the validity of the main conclusions as now stated more clearly.

Referee #1: I support presentation of the work with a new title and rephrasing of the findings in line with the most likely explanation and caveats discussed here.

Response: We thank the reviewer and updated the title and rephrased the findings.

Referee #1: Minor comment: is "drug exposure" in EV3 is for just Mli-2 or for both MLI-2 and nocodazole?

Response: The figure has been updated to remove the term "Drug exposure", which was confusing, and the legend updated to better indicate exposures were both for MLI2- and nocodazole.

Sincerely,

Tuyana Malankhanova and

Andrew West

Prof. Andrew West
Duke University
Pharmacology and Neurology
3 Genome Court
Durham, NC 27710
United States

Dear Andy,

Thank you for submitting your revised manuscript. I have now looked at everything and all is fine. Therefore, I am very pleased to accept your manuscript for publication in EMBO Reports.

Congratulations on a nice work!

Before we can export your manuscript to our publisher, I need your input on one final point. I note that the synopsis image you provided is comprised of immunofluorescent images of pRab10/LAMP1/Tubulin stainings. However, this image should provide a rapid overview of the question addressed in the study (550 (width) x 300-600 (height)). You can send me the synopsis image by responding to this email. Thank you.

Kind regards,

Deniz

--

Deniz Senyilmaz Tiebe, PhD
Senior Scientific Editor
EMBO Reports
